# A multi-objective based clustering for inferring BCR clonal lineages from high-throughput B cell repertoire data

**Nika Abdollahi**[1], **Lucile Jeusset**[1,2], **Anne Langlois De Septenville**[2], **Hugues Ripoche**[1], **Frédéric Davi**[2], **Juliana Silva Bernardes**[1]*

**1** Sorbonne Université, CNRS, UMR 7238, Laboratoire de Biologie Computationnelle et Quantitative, Paris, France, **2** Sorbonne Université, AP-HP, Hôpital Pitié-Salpêtrière, UMR_S 1138 Department of Hematology, Paris, France

* juliana.silva_bernardes@sorbonne-universite.fr

**Data Availability Statement:** Data and the source code are freely available at github.com/julibinho/MobiLLe.

## Abstract

The adaptive B cell response is driven by the expansion, somatic hypermutation, and selection of B cell clonal lineages. A high number of clonal lineages in a B cell population indicates a highly diverse repertoire, while clonal size distribution and sequence diversity reflect antigen selective pressure. Identifying clonal lineages is fundamental to many repertoire studies, including repertoire comparisons, clonal tracking, and statistical analysis. Several methods have been developed to group sequences from high-throughput B cell repertoire data. Current methods use clustering algorithms to group clonally-related sequences based on their similarities or distances. Such approaches create groups by optimizing a single objective that typically minimizes intra-clonal distances. However, optimizing several objective functions can be advantageous and boost the algorithm convergence rate. Here we propose MobiLLe, a new method based on multi-objective clustering. Our approach requires V(D)J annotations to obtain the initial groups and iteratively applies two objective functions that optimize cohesion and separation within clonal lineages simultaneously. We show that our method greatly improves clonal lineage grouping on simulated benchmarks with varied mutation rates compared to other tools. When applied to experimental repertoires generated from high-throughput sequencing, its clustering results are comparable to the most performing tools and can reproduce the results of previous publications. The method based on multi-objective clustering can accurately identify clonally-related antibody sequences and presents the lowest running time among state-of-art tools. All these features constitute an attractive option for repertoire analysis, particularly in the clinical context. MobiLLe can potentially help unravel the mechanisms involved in developing and evolving B cell malignancies.

## Author summary

High-throughput sequencing can produce a large set of sequences and has profoundly changed our ability to study immune repertoires, particularly B cell receptor sequences. An important application is the analysis of the clonal lineage composition of B cell

**Funding:** This work has been supported by Sorbonne Universite grants: NA - "2016 Programme Doctoral de Cancérologie" (https://www.edite-de-paris.fr) LJ,JSB, FD - SIRIC CURAMUS (https://curamus-cancer.fr/en/) The funders had no role in study design, data collection and analysis, decision to publish, or preparation of the manuscript.

**Competing interests:** The authors have declared that no competing interests exist.

populations; it is the starting point of many immune repertoire studies, for instance, to differentiate between healthy individuals and those with lymphoid malignancies or other diseases. Several computational methods have been developed to identify clonal lineages from a set of B cell receptor sequences. Most of them apply clustering algorithms and optimize a single objective function that typically minimizes intra-clonal distances. However, optimizing several objective functions in parallel can benefit and increase the clustering performance and efficiency. We propose MobiLLe, the first multi-objective clonal lineage grouping method, which simultaneously optimizes two objective functions for minimizing intra-clonal diversity and maximizing inter-clonal differences. Our approach greatly improved clonal grouping on simulated benchmarks and performed comparably to the most powerful and recent methods on experimental samples. MobiLLe is computationally more efficient than existing tools and does not require any training process or hyper-parameter optimization. It can easily manage large-scale experimental repertoires, providing useful plots to help researchers detect clonally-related sequences in high-throughput B cell repertoire data.

This is a *PLOS Computational Biology* Methods paper.

## 1 Introduction

B and T lymphocytes are the major cellular components of the adaptive immune system. They are defense cells that can distinguish between self and non-self antigens, promoting the destruction of non-self antigen-bearing cells such as pathogens or tumor cells. Lymphocytes can recognize or directly bind to antigens via their membrane-bound specific receptors: B cell and T cell receptors (BCR/TCR). These receptors are composed of a recognition unit, the immunoglobulin (IG) in the case of the BCR, and a signaling unit, CD79, and CD3 for B and T cells, respectively. The recognition units are heterodimers consisting of two heavy (IGH), two light (IGL) chains for IG, and two chains for TCR, alpha-beta, much more often than gamma-delta. Each chain in TCR/BCR has an N-terminal variable region (V), a constant region (C), a transmembrane region, and a short C-terminal cytoplasmic tail. BCRs and TCRs belong to a unique class of proteins whose coding sequences are assembled through a complex genetic mechanism known as V(D)J recombination [1], occurring during the early stages of B and T cell maturation. For TCR-beta, delta, and BCR-heavy chains, three sets of genes encode the variable region: variable (V), diversity (D), and joining (J). These genes are naturally separated on the genome, but they are brought together during V(D)J recombination, which randomly selects one of each of three types of genes from a pool of many germline variants and joins them [2] to form the V (variable) region of BCR and TCRs. Joining is imprecise as nucleotides are randomly deleted and inserted in the V-D (N1) and D-J (N2) junctions, thereby further enhancing the combinatorial diversity by considerable junctional diversity. The N1-D-N2 region is at the center of the so-called third complementarity determining region (in short, CDR3) and has the highest variability within the V regions. Conversely, IG light chains, and TCR alpha/gamma chains are composed of only two types of genes: variable (V) and joining (J), and have shorter and somewhat less diverse CDR3.

In addition to V(D)J recombination, B cells but not T cells can undergo further diversification in their antigen receptors after antigen encounter by introducing a high number of nucleotide mutations in their variable regions by a process called somatic hypermutation (SHM) [3]. These mutations affect the affinity of the BCRs for their cognate antigens, and B cells expressing BCR with enhanced affinity are positively selected, resulting in an antigen-driven clonal selection. These genetic processes create a functionally diverse and dynamic set of B and T cells, equipped with an extremely diversified immune repertoire of antigen receptors, with an estimated $10^{12}$ different antigen specificities.

High-throughput sequencing of antigen receptors now offers unprecedented opportunities to evaluate the composition and immune repertoire diversity of various lymphocyte populations. As a first step, a common approach is to assess and quantify the clonality of antigen receptor variable regions by grouping identical sequences (often referred to as a clonotype). In BCR repertoires, one can also group sequences with the same V(D)J genes (or alleles), and identical CDR3 amino acid sequence [4]. Such a group of sequences forms a so-called subclone [5, 6] and sequences within this group belong essentially to the same cellular clone and derive from a common ancestor. As mentioned above, upon antigen activation, B cells undergo rapid proliferation and further diversification of their BCR sequences by SHM, introducing nucleotide substitutions into the BCR variable regions. This occurs mainly in highly specialized structures, the germinal centers of secondary lymphoid organs, where a selection process called affinity maturation operates. B cells for which SHM produced BCR with higher affinity for their cognate antigen expand, while those with a lower affinity are eliminated, thereby contributing to the affinity maturation of the B lymphocytes. As a result, antigen-specific B cell lineages with increased BCR affinity are produced. Therefore a (theoretical) B cell lineage includes the unmutated ancestor and all mutated variants. S1 Fig illustrates the different levels of grouping BCR repertoire sequences.

As TCR sequences do not undergo SHM; it is easier to identify clonally-related TCRs once identical sequences form a clonal lineage. Here we focus on the BCR clonal lineage grouping task, which has scientific and clinical importance in physiological and pathological contexts. For instance, identifying clonal lineage in BCR repertoires (BCR clonal lineage grouping) is generally the starting point for several studies involving distinct clinical contexts like autoimmune diseases [7], allergy [8], cancer [5], ageing [9], and immune responses to infections [10, 11]. Moreover, it is also a commonly used way to distinguish clonal (tumoral) from non-clonal (non-tumoral) cell populations in case of suspicion of lymphoid malignancies [12]. Extremely varied repertoires are called polyclonal and are generally observed in healthy individuals. In contrast, individuals diagnosed with lymphoproliferative diseases such as leukemia or lymphoma typically present monoclonal repertoires in which there is one highly expanded clonal lineage. Between these two extreme situations, the immune repertoire can display unique or multiple relatively minor clonal expansions reflecting various perturbations of the immune homeostasis, giving rise to an oligoclonal immune repertoire.

Advanced sequencing techniques along with reduction of costs have encouraged researchers to conduct deeper investigations of BCR repertoires. A huge amount of sequence data can help estimate the diversity of BCR repertoires, detect B cell malignancies, and understand the antigen-driven evolution of B cells, among others. In particular, identifying clonal lineage from high-throughput B cell repertoire data can help reconstruct cell lineage and unravel inter/intra clonal repertoire diversity. Several computational methods for BCR clonal lineage grouping have been developed, which generally employ clustering algorithms to infer clonal relationships [13–15]. Most methods perform clonal lineage grouping in two main steps. First, sequences with the same IGHV and IGHJ genes, and CDR3 of the same length, are grouped. Second, the sequences within each group are clustered according to some sequence-based

distance. Any standard clustering approach can be applied, such as hierarchical [16], spectral [14] or agglomerative clustering [13]. An alternative to these clustering approaches is to construct a lineage tree and cut it to create sub-trees, or clonal lineages [17, 18]. All previous methods focus mainly on minimizing intra-clonal distances; they are based on only one criterion, reflecting a single measure of the partitioning quality. Such a single measure might not capture the different characteristics of datasets, whereas a multi-objective approach might be more appropriate.

Here we propose MobiLLe, a Multi-Objective Based clustering for Inferring BCR clonal lineage from high-throughput B ceLL rEpertoire data. MobiLLe requires IGHV and IGHJ gene annotations and a fixed CDR3 amino acid identity threshold to form initial groups. Next, we merge groups if this minimizes intra-clonal diversity and maximizes inter-clonal differences. Thus, MobiLLe optimizes two objective functions in parallel that continually evaluate groups' consistency until no improvement is observed in their cohesion or separation. By minimizing intra-clonal diversity, we improve each group's cohesion, which measures how similar sequences are within the clonal lineage. On the other hand, we improve the separation among distinct clonal lineages by maximizing the inter-clonal differences. We show that our approach greatly improves BCR clonal (lineage) grouping on simulated benchmarks and performs comparably to the most powerful and recent methods on experimental BCR repertoires. MobiLLe produces reliable partitioning when existing clonal (lineage) methods fail, being very stable even on higher sequence mutation rates. When applied to experimental BCR repertoires, it inferred similar clonal distributions to the most performing methods and was able to reproduce the results of recent publications. However, MobiLLe has high scalability, low runtime, and minimal memory requirement.

## 2 Methods

### 2.1 MobiLLe

Multi-objective clustering (MOC) decomposes a dataset into related groups, maximizing multiple objectives in parallel. Several frameworks exist to implement MOC, MobiLLe relies on multi-run clustering, where a clustering algorithm runs multiple times to optimize different objectives that capture a compound fitness function [19]. MobiLLe proceeds through two main steps: pre-clustering and refinement. Fig 1 shows its flowchart, and Algorithm 1 and 2 the pseudo-code for the refinement step.

**2.1.1 Pre-clustering.** The pre-clustering step aims at grouping similar sequences to form initial clonal lineage groups that can be refined later. Note that MobiLLe takes only IGH sequences as input since they are more diverse than IGL chains, providing a reliable signature for immune repertoire studies [20]. First, BCR sequences are annotated to identify their IGHV and IGHJ genes (and alleles) and locate their CDR3 regions. For this purpose, we used IMGT/ HighV-QUEST [21], but in principle, any V(D)J annotation software could be used since MobiLLe accepts input data in AIRR format [22]. Sequences with the same IGHV and IGHJ genes and the same CDR3 sequence length are then grouped. Finally, we separate sequences with less than $s$% of CDR3 amino acid identity (by default $s$ is 70%), see the "pre-clustering" panel in Fig 1.

**2.1.2 Clustering refinement.** In this step, we iteratively refine clonal lineage groups until we reach the minimum values for intra-clonal distances and the maximum values for inter-clonal distances. The algorithm described in 1 takes the set of initial groups $C$ as input; generated during the pre-clustering step. For each group $K \in C$ and each sequence $i \in k$ it computes two distances: $a_i$ (intra-clonal) and $b_i$ (inter-clonal). Such distances measure the cohesion/separation within detected groups; they were initially introduced to compute the Silhouette [23].

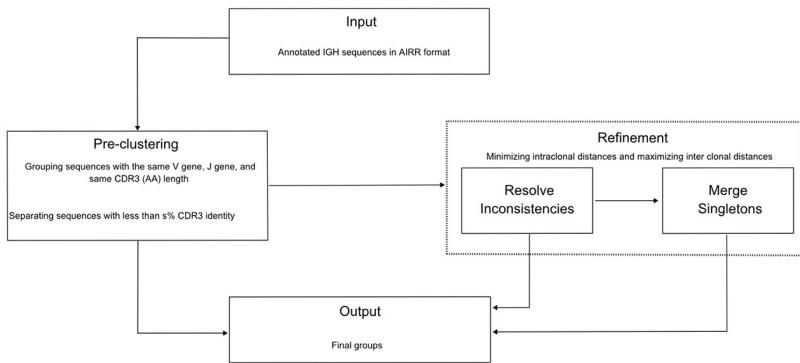

**Fig 1. Flowchart of MobiLLe.** MobiLLe requires IGH annotated sequences (IGHV, IGHJ, and CDR3 region were previously identified) to form initial clusters (pre-clustering panel), we first group sequences with the same IGHV, IGHJ, and same CDR3 (AA) length; then, we separate sequences with less than s% CDR3 identity (default 70%). Refinement has two steps: 'resolve inconsistencies' and 'merge singletons.' The first one detects and resolves inconsistencies until no improvement is observed in cluster cohesion or separation. The second one tries to merge singletons into higher-density clusters to improve their uniformity. Final groups (output) represent clonal lineages with low intra-clonal diversity, high inter-clonal diversity, and a minimum number of singletons.

$a_i$ is the average distance between the sequence $i$ and any other sequence in the same group $K$; $b_i$ is the smallest average distance of $i$ to all sequences in any other group $L$. In a well-separated cluster, $a_i$ is smaller than $b_i$, thus, if for a given sequence $a_i$ is higher than $b_i$, it indicates an inconsistency, showing that the clusters $K$ and $L$ should be merged. Note that $a_i$ needs to be recomputed for all sequences in the merged cluster, and $b_i$ for all sequences. Consequently, each cluster merging launches a new iteration of the algorithm, and it stops if no merging is observed in the previous iteration or after a predefined number of iterations.

Certainly, the distance metric $d(i, j)$ (between sequences $i$ and $j$) plays an important role when computing $a_i$ and $b_i$. Distances based on sequence similarity of the whole sequences can be inaccurate since different IGHV, and IGHJ genes can present considerable similarities. Moreover, CDR3 regions are shorter than IGHV/IGHJ genes, and a normalized distance should be more appropriate. Therefore, we defined a composed distance that splits sequences into three parts, IGHV, IGHJ, and CDR3 region, and computed a distinct distance of each part separately. The final distance $d(i, j)$ is the ponderate mean of these three distances and is defined by the equation:

$$d(i,j) = \frac{\alpha * d_{ij}^{V} + \beta * d_{ij}^{CDR3} + \lambda * d_{ij}^{J}}{\alpha + \beta + \lambda}, \tag{1}$$

where $d_{ij}^{V}$ is the IGHV distance, $d_{ij}^{CDR3}$ is the CDR3 distance, $d_{ij}^{J}$ is the IGHJ distance, and $\alpha$, $\beta$, and $\lambda$ are coefficients. Note that when $\alpha = \beta = \lambda$ the arithmetic mean is computed. There are several ways to compute $d_{ij}^{V}$, $d_{ij}^{CDR3}$ and $d_{ij}^{J}$ distances; we have implemented four different distance types:

1. the binary distance is based on gene identifications; it is 0 if $i$ and $j$ were annotated with the same IGHV/IGHJ gene or 1, otherwise. For CDR3 $d_{ij}^{CDR3}$ is 0 when $i$ and $j$ CDR3 amino acid sequences are identical, and 1, otherwise,

2. the normalized Levenshtein distance [24] computes the minimum number of single-character editions (insertions, deletions, or substitutions) required to transform $i$ into $j$, to normalize it, we divided by the length of the longer sequence;

3. the k-mer based distance [25] is based on the fractional of common k-mers present on both sequences; it is zero if $i$ and $j$ are identical, and one if they do not share any k-mer;

4. the GIANA [26] is an isometric distance that encodes amino acid sequences into numeric vectors through a series of unitary transformations. After encoding, sequences are projected to high-dimensional space, allowing a rapid distance calculation between points in the Euclidean space.

**Algorithm 1**: Clustering refinement

```
Require: C {initial groups}
 repeat
    stop ← true
    for all k ∈ C do
      if |k| > 1 then
        for all i ∈ k do
          a_i ← (1/|k|-1) ∑_{j∈k} d(i,j)
          b_i ← min_{l≠k}∀_{j∈l} d(i, j)
          l = argmin_l ∀_{j∈l} d(i, j)
          if a_i > b_i then
            merge clusters k and l
            stop ← false
          end if
        end for
      end if
    end for
 until not stop
```

The last step of the refinement tries to reduce the number of singletons. This is an important step because once singletons are formed, they cannot be merged to higher density clusters, just using Algorithm 1, since their intra-clonal distance $a_i$ is zero, and it is smaller than any other inter-clonal distance. Thus, we defined the Algorithm 2 that considers every singleton $k$, and tries to merge it to its closest neighbor $l$ if the merging does not substantially decrease cluster uniformity [27]. More precisely, we merge $k$ and $l$ if $|unif(l + k) - unif(l)| < \delta$. The cluster uniformity measures the variability of a cluster, and smaller values represent a more uniform data distribution. The following equations compute cluster uniformity:

$$\text{unif(k)} = \begin{cases} \dfrac{\sum_{i=1}^{n} \text{local\_den}(i) - \text{avg\_den}(k)}{\text{avg\_den}(k)} & , i \in k, n > 1 \\ 0 & , n = 1 \end{cases} \tag{2}$$

$$\text{local\_den}(i) = min\{d(i,j)\}, \forall i \in k, i \neq j \tag{3}$$

$$\text{avg\_den}(k) = \frac{\sum_{i=1}^{n} \text{local\_den}(i)}{n} \forall i \in k \tag{4}$$

where $n$ is the total number of sequences in the cluster $k$, and d(i,j) is the composed distance defined above.

**Algorithm 2**: Merging singletons

```
Require: C {set of singletons}
  for all k ∈ C do
    i ← k[0] {Take the single sequence of k}
    l = argmin_{l≠k}∀_{j∈l} d(i, j) {Find the closest neighbor of k}
    if |unif(l + k) - unif(l)| < δ then
      merge clusters k and l
```

```
        end if
      end for
```

## 2.2 Data sets

To evaluate MobiLLe and compare it with existing BCR clonal lineage grouping methods, we used three types of BCR repertoire data: simulated, artificial, and experimental.

**2.2.1 Simulated repertoires.**   IGH simulated sequences are largely used to evaluate clonal lineage grouping methods [13, 14, 28]. Some repertoire simulators have been proposed as part of such tools, but an independent B cell repertoire simulator that could produce different types of IGH repertoires (clonal, non-clonal), to the best of our knowledge, does not exist. In order to create simulated repertoires, we adapted GCTree [29], a B cell lineage simulator. We ran GCTree several times to produce independent B cell lineages, which were then assembled in a single repertoire.

To produce a B cell lineage, GCTree randomly selects IGHV, IGHD, and IGHJ germline genes from the IMGT database [4], then nucleotide(s) can be added to or removed from the IGHV-IGHD and IGHD-IGHJ junction regions. Next, a branching process is performed, and point mutations are included in the descendants to simulate SHM. For the branching, GCTree uses an arbitrary offspring distribution that does not require an explicit bounding. Instead, it uses a Poisson distribution with parameter $\lambda$ to estimate each node's expected number of off-spring. SHM are simulated by a sequence-dependent process, where mutations are preferentially introduced within certain hot-spot motifs. GCtree uses the 5-mer context model [30] to compute the mutability $\mu_1, \ldots, \mu_i \ldots, \mu_l$ for each residue $i$ of a sequence of length $l$. The mutability of the whole sequence $\mu_0$, is then computed by averaging the mutability of its residues: $\mu_0 = \frac{1}{l}\sum_{i=1}^{l}\mu_i$. To determine the number of mutations $m$ to be introduced in each mutant offspring sequence, GCTree also uses a Poisson distribution with parameter $\lambda_0$, $m$ is then computed as $Pois(\mu_0\lambda_0)$; note that more mutable sequences (higher $\mu_0$) tend to accumulate more point mutations.

Basically, the GCTree simulator has two main parameters to be set: $\lambda$, to estimate the expected number of offspring of each node, and $\lambda_0$, to determine the number of point mutations in mutant offspring sequences. We kept $\lambda$ as the default value (e.g., 2), but we varied $\lambda_0$ to produce simulations with different mutation rates. We experimented with four values {0.16, 0.26, 0.36, 0.46}, where 0.26 is the default value. Note that higher $\lambda_0$ values produce more divergent B cell lineages. For each $\lambda_0$ setting, we simulated three types of repertoires: monoclonal, oligoclonal, and polyclonal, obtaining 12 different benchmarks. Each repertoire type's initial clonal size setting is shown in S1 Table. Since we only kept productive sequences (without stop codons), the simulated repertoires, Table 1, can contain fewer sequences than shown in S1 Table.

**2.2.2 Artificial monoclonal repertoires.**   Gold standard experimental data, where truly clonal relationships are known with certainty, are challenging to obtain. In order to create experimental-based benchmarks, we constructed artificial monoclonal repertoires by combining sequences from a pure B cell lineage and a polyclonal repertoire. Our goal was to determine if clonal lineage grouping methods could separate sequences from these two sources. To form a benchmark, we considered 10000 sequences, where 10% of them were sampled from the pure lineage and 90% from the polyclonal background. Since we know the truly clonally-related sequences in each benchmark, we could compare the different tools for determining their grouping differences. We created three artificial monoclonal repertoires with three different pure B cell lineages, each having a specific V(D)J rearrangement. The pure B cell lineage of the artificial monoclonal benchmark AMR1 was annotated with IGHV1–69*01/IGHJ6*03

**Table 1. Simulated repertoire properties.**

| Label | $\lambda_0$ | Clonality | number of sequences | number of clonal lineages |
|-------|-------------|-----------|---------------------|---------------------------|
| M16 | 0.16 | Monoclonal | 958 | 34 |
| O16 | | Oligoclonal | 1014 | 43 |
| P16 | | Polyclonal | 968 | 44 |
| M26 | 0.26 | Monoclonal | 659 | 33 |
| O26 | | Oligoclonal | 958 | 43 |
| P26 | | Polyclonal | 964 | 44 |
| M36 | 0.36 | Monoclonal | 924 | 35 |
| 036 | | Oligoclonal | 991 | 40 |
| P36 | | Polyclonal | 897 | 42 |
| M46 | 0.46 | Monoclonal | 952 | 35 |
| O46 | | Oligoclonal | 1016 | 43 |
| P46 | | Polyclonal | 952 | 43 |

genes, AMR2 with IGHV3–48*02/IGHJ4*02 rearrangements, and the AMR3 with the IGHV3–15*01/IGHJ6*02 rearrangements. We used IMGT/HighV-QUEST [21] as V(D)J assignment tool. AMR1, AMR2, and AMR3 pure B cell lineage were sampled from 22747, 20371, and 23665 sequences, while polyclonal backgrounds were from 136977 sequences. S2–S4 Figs show IGHV/J gene usage distribution of the polyclonal background compared to AMR1, AMR2, and AMR3 sequences, respectively.

The pure B cell lineages and polyclonal backgrounds are human peripheral blood mononuclear cells obtained during routine diagnostic procedures at Pitié-Salpêtrière hospital (Paris-France). DNA sequences were obtained by polymerase chain amplification of IGH-VDJ rearrangements followed by paired-end sequencing on an Illumina MiSeq platform. We obtained one "Read 1" and "Read 2" FASTQ files for each sample, which were then merged by the PEAR software [31]. The merged FASTQ files were converted to FASTA format with seqtk (https://github.com/lh3/seqtk).

**2.2.3 Experimental repertoires.** In order to evaluate our approach on realistically-sized repertoires, we used a dataset produced at the Pitié-Salpêtrière hospital (Paris-France), and two public datasets from IReceptor repository [32], totalizing 26 experimental repertoires.

The first dataset contains nine samples of human peripheral blood mononuclear cells collected during routine diagnostic procedures at Pitié-Salpêtrière hospital. We picked three samples from these repertoires to systematically compare MobiLLe's results and existing clonal lineage grouping methods. DNA sequences were obtained and assembled as described in Section 2.2.2 Table 2 shows the number of reads (sequences), the number of unique sequences (clonotypes), and the clonality status defined by standard PCR amplification and capillary electrophoresis of amplicons (Genescan analysis) [33]. S5 Fig shows the output plot of Genescan profiles for each experimental repertoire.

The second dataset contains five experimental repertoires selected from a previous work that performed high-throughput sequencing to characterize the B cell populations in several lymphoproliferative diseases [34]. Samples come from different tissues such as blood, lymph nodes, liver, and bone marrow. From this study, we selected one healthy donor (HD) and four patients diagnosed with different lymphoproliferative diseases. Sequence data were obtained through the sequencing platform 454 GS-FLX. Table 3 summarizes the main properties of selected repertoires, for more details report to [34].

**Table 2. Properties of nine experimental repertoires.** For each sample, we show clonality status, the individual label, the total number of sequences and unique sequences (clonotypes).

| Clonality | label | Total of Sequences | Total of unique sequences |
|---|---|---|---|
| Monoclonal | I1 | 33599 | 22181 |
| | I2 | 65853 | 26431 |
| | I3 | 61990 | 19949 |
| Oligoclonal | I4 | 73888 | 53093 |
| | I5 | 294203 | 219006 |
| | I6 | 140026 | 84070 |
| Polyclonal | I7 | 57076 | 40673 |
| | I8 | 70050 | 61379 |
| | I9 | 162742 | 104923 |

The third dataset contains 12 repertoires selected from a previous work that conducted immune profiling studies of several individuals with different trajectories of SARS-CoV-2 infection and COVID-19 (moderate and severe) and compared them to healthy donors [35]. Samples were collected from peripheral blood mononuclear cells, and sequencing was performed with the Illumina MiSeq platform; Table 4 summarizes the properties of 12 considered repertoires; for more details, report to the original publication.

## 2.3 Performance evaluation

**2.3.1 Clustering accuracy.** When clonal assignments are known, one can quantitatively assess clonal lineage grouping algorithms' performance by measuring their ability to identify clonally-related sequences. For that, we applied classical measures such as precision and recall for comparing the inferred clusters (clonal lineages) to the true ones. Consistently, we also computed the F-score (FS), the harmonic mean of precision and recall; it is an aggregate measure of the inferred cluster's quality. Precision and recall require three disjoint sets, which are: true positive (TP), false-positive (FP), and false-negative (FN). From these, we compute precision $p = \frac{|TP|}{|TP|+|FP|}$, recall $r = \frac{|TP|}{|TP|+|FN|}$, and FS $= \frac{2*p*r}{p+r}$. The values of these three metrics are in the interval [0, 1], being one the best and zero the worst performance. Certainly, the way TP, FP, and FN are computed will affect the accuracy measures. Depending on the grouping level, there are at least two ways to compute these values: *pairwise* and *closeness*.

**Table 3. Properties of patients and healthy donor repertoires.**

| Label | Description | Sample Type | Clonality Assay | Nb sequences |
|---|---|---|---|---|
| HD | Healthy donor | Blood | non-clonal | 17641 |
| P3-FL-SLL | Patient 3, FL, SLL | Lymph node | clonal | 13875 |
| P4-S/CLL | Patient 4, SLL, CLL | Blood | clonal | 5017 |
| P5-BM | Patient 5, PTLD | Bone marrow | clonal | 7635 |
| P5-L | Patient 5, PTLD, DLBCL | liver | clonal | 3345 |

The clinical clonality assay was obtained by standard PCR amplification and capillary electrophoresis of PCR products. Abbreviations: Nb: number; Blood: peripheral blood mononuclear cells; Lymph node: formalin-fixed paraffin-embedded lymph node tissue; Liver: formalin-fixed paraffin-embedded liver tissue; CLL/SLL chronic lymphocytic leukemia/small lymphocytic lymphoma; FL follicular lymphoma; PTLD post-transplant lymphoproliferative disease; DLBCL diffuse large B cell lymphoma [source [34]].

**Table 4. Clinical and repertoire characteristics of healthy donors and patients with moderate/severe COVID-19.**

| Label | Description | Age Bracket | PD | VMD | Comorbidites | Nb sequences |
|---|---|---|---|---|---|---|
| H3 | Healthy donor | 24–61 | N | N | | 241126 |
| H4 | Healthy donor | 24–61 | N | N | | 238679 |
| H8 | Healthy donor | 24–61 | N | N | | 125567 |
| M5 | Moderate | 25–30 | N | N | | 308242 |
| M6 | Moderate | 61–65 | Y | Y | | 280459 |
| M7 | Moderate | 41–45 | N | Y | | 163032 |
| S20 | Severe | 71–75 | Y | Y | Autoimmune disease | 268602 |
| S21 | Severe | 71–75 | N | Y | Bacteremia gram negative | 58189 |
| S22 | Severe | 76–80 | N | Y | | 57735 |
| S23 | Severe | 81–85 | N | Y | | 87050 |
| S24 | Severe | 51–55 | N | Y | B cell lymphoma | 285852 |
| S26 | Severe | 71–75 | N | Y | | 72443 |

Abbreviations: PD: Pulmonary Disorder (asthma, sarcoidosis, chronic obstructive pulmonary, disease or interstitial lung disease); VMD: Vascular and metabolic disorder (obesity, cardiovascular disease, hypertension, diabetes mellitus and hyperlipidemia) Nb: number; Y, yes. N, no; [source [35]].

The *pairwise* procedure considers the binary clustering task and focuses on the relationship between each pair of sequences. A pair of sequences is counted as: TP if the sequences are found together in both 'true' and 'inferred' clusters; FP if the sequences are found separated in the true but together in the inferred cluster; FN if the pair is found together in the true but separated in the inferred cluster, see an illustration in S6(A) Fig.

The *closeness* procedure evaluates clonal compositions and the repertoire structure. For that, we first identified the best correspondence between inferred clonal lineages and correct clonal assignments. Then, we associated clonal lineage pairs sharing the maximum of common sequences. For each associated pair $i$, considering $I_i$ inferred and $T_i$ the correct clonal assignment, we computed $TP_i$ as the intersection between the two sets ($I_i \cap T_i$), $FP_i$ as the difference between inferred and the correct clonal assignment ($I_i \backslash T_i$), and $FN_i$ as the difference between the correct clonal assignment and inferred group ($T_i \backslash I_i$). Finally, we computed $TP = \cup_i^n TP_i$, $FP = \cup_i^n FP_i$ and $FN = \cup_i^n FN_i$, where $n$ is the number of associated clonal lineage pairs; see an illustration in S6(B) Fig.

**2.3.2 Comparison of clonal distributions.** In order to compare clonal lineages obtained by different tools, we have defined five "events" that describe the differences between each pair of clonal distributions. For this, we labelled clusters in a clonal distribution $d_2$ by comparing them with clusters in a distribution $d_1$. These events are represented in Fig 2, and can be interpreted as follows:

1. identical: clusters in both distributions are identical, they contain the same set of sequences (Fig 2A),

2. join: when sequences of different clusters in $d_1$ were joined in the same cluster in $d_2$ (Fig 2B),

3. split: when sequences of a cluster in $d_1$ were divided into several clusters in $d_2$ (Fig 2C),

4. Mix: when a mixture of the three above events occurs. For instance, in Fig 2D, we observed two events, "split" ($C_8$ and $C_9$) and "join" ($C_{10}$, $C_{11}$ and $C_{12}$),

5. Not found: when a cluster in $d_2$ was not found among clusters in $d_1$. For instance, in Fig 2E, cluster $C_{13}$ was not present in $d_1$.

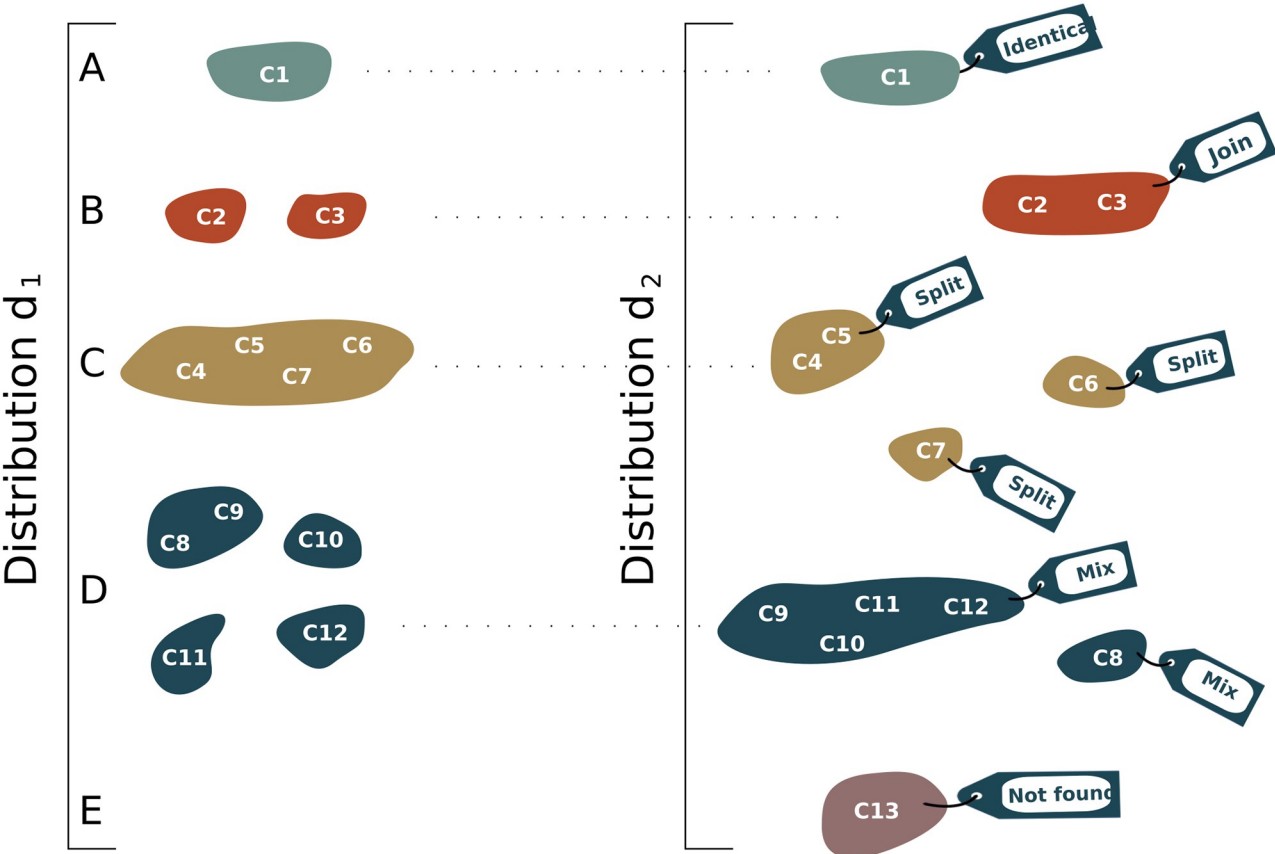

**Fig 2. Clonal distribution comparisons.** Five "events" describe the differences between two clonal distributions ($d_1$ and $d_2$). The identical event counts the number of identical clonal lineages found in both distributions (a). The "join" event reports the number of clonal lineages in $d_1$ found merged in $d_2$ (b), while the split counts the number of clonal lineages in $d_1$ found separated in $d_2$ (c). The "mix" event is a mixture of these two later events (d), while "not found" reports the number of clonal lineages in $d_2$ not found in $d_1$ (e).

## 2.4 Clonal lineage grouping tools considered for comparisons

MobiLLe groups within a cluster those sequences that might belong to a B cell lineage. Thus, we chose four tools that employ the same clonal grouping definition. Each method has its own set of characteristics concerning the underlying algorithm, prior information, and produced outputs. Here we briefly describe them; report to original publications for more details.

**2.4.1 BRILIA.** B cell Repertoire Inductive Lineage and Immunosequence Annotator (BRILIA) builds up lineage tree reconstruction, clonal grouping, and V(D)J annotation into a single algorithm [18]. From a collection of IGH sequences, BRILIA first provides initial V(D)J gene identification according to the IMGT database [4]. Then, it groups sequences with the same IGHV and IGHJ gene subgroups and the same CDR3 sequence length. It proceeds by reconstructing lineage trees that will determine groups of clonally-related sequences. For that, it determines parent-child sequence relationships within each group for further reconstructing lineage trees. Evolutionary relationships are based on an adjusted hamming distance that penalizes dissimilarities in the N regions. Next, BRILIA determines the root of each obtained tree as the sequence involved in a cyclic dependency having the smallest distance to all other sequences in that cluster. Finally, a clone is a group of sequences sharing a common root sequence.

**2.4.2 Partis.** Partis [13] uses hidden Markov model (HMM) to represent V(D)J rearrangement events [36]. An HMM is a probabilistic model, where the modeled system is assumed to be a Markov process with hidden states and unknown parameters. Each hidden state emits a symbol representing an elementary unit of the modeled data; for example, in BCR sequences, the hidden states represent either gene positions or N-region (addition or deletion) nucleotides. Thus, the HMM states represent nucleotides of each IGHV, IGHD, and IGHJ gene. The emission probabilities incorporate the probability of SHM at each nucleotide, and transition probabilities represent the probability of moving from one state to another. The HMM parameters (emission and transition probabilities) are estimated from a large panel of available sequences. Once the model is trained, BCR sequences are annotated by computing the Viterbi path through the HMM and finding the maximum-likelihood annotation. After V(D)J assignment, Partis applies its clonal grouping strategy. First, it creates initial clonal lineages of sequences sharing the same IGHV and IGHJ genes and the same CDR3 length. Then, it applies an agglomerative clustering algorithm to merge clusters that maximize the likelihood ratio that could indicate that two clusters derive from the same rearrangement events.

**2.4.3 SCOPe.** SCOPe requires V(D)J annotation before clonal grouping, and tools such as IMGT/HighV-QUEST [21], or IgBlast [37] can be used. To identify a clone, SCOPe applies a spectral clustering method with an adaptive threshold to determine the local sequence neighborhood; meaning that it does not require a fixed threshold for detecting clonally-related sequences. Given a set of IGH sequences, SCOPe first divides them into groups with the same IGHV gene, IGHJ gene, and junction length. Then, it computes the similarity matrix for each group considering the hamming distance between junction regions of each pair of sequences within the group. Next, it generates a fully connected graph from the data points and performs local scaling to determine the local neighborhood. Based on the graph, SCOPe builds an adjacency matrix and creates a Laplacian graph. The eigenvalues of such a graph can then be used to find the best number of clusters, and the eigenvectors can be used to find the actual cluster labels. Finally, SCOPe performs k-means clustering on the eigenvectors to get the labels (clone) for each node (sequence).

**2.4.4 SONAR.** For SONAR (Software for the Ontogenic aNalysis of Antibody Repertoire) [17], a clonal group contains all IG reads that share a common ancestor. This tool focuses further on seeded lineage assignment, where the sequences of one or more known antibodies are used as seeds to find all sequences in the dataset from the same lineage while leaving the rest of the sequences unclassified. In addition, it can perform "unseeded lineage assignment," which consists of classifying sequences into component lineages without any additional information. In order to perform an unseeded lineage assignment, SONAR separates sequences based on their assigned IGHV and IGHJ genes. The sequences in each group are then clustered based on their CDR3 nucleotide identity (by default, 90% of CDR3 sequence), using the UCLUST algorithm in USEARCH [38]. Eventually, each clone is identified as a distinct unseeded lineage.

## 3 Results

We first evaluated and optimized MobiLLe on simulated repertoires, where clonally related sequences are well defined for all clonal lineage. Next, we compared MobiLLe with state-of-art methods on simulated, artificial, and experimental repertoires. Finally, we demonstrated the usefulness of the MobiLLe tool by applying it to experimental repertoires obtained in various clinical situations, including chronic lymphocytic leukemia and COVID-19. All datasets are described in Section 2.2.

### 3.1 Parameter optimization

In order to check the influence of the MobiLLe parameter setting on the clonal grouping accuracy, we exhaustively varied the pre-clustering and refinement parameters. We ranged the pre-clustering $s$ threshold from 50% to 90% with a step of 10, totalizing five different possibilities. For refinement parameters, we varied $d_{ij}^V$, $d_{ij}^{CDR3}$, and $d_{ij}^J$ distances with four different values: binary, Levenshtein, GIANA and k-mer-based distance, totalizing $4^3 = 64$ different possibilities. We also varied the coefficients $\alpha$, $\beta$, and $\lambda$ to compute different pondered distance means, totalizing $C_3^6 + 1 = 7$ combinations. Finally, we ran MobiLLe with/without 'merge singleton' algorithm with $\delta = 0.05$, see Section 2.1.2. All these variations produced $4^{3*}5^*7^*2 = 4480$ different configurations. Thus, we ran MobiLLe 53760 times (4480 x 12) on 12 generated benchmarks that simulated three repertoire types with different SHM rates.

The simulated repertoires were produced with GCTree [29], a B cell lineage simulator, which randomly selects germline sequences for generating the unmutated common ancestor of each lineage and then introduces point mutations at hot-spot positions. We ran GCTree several times to create a collection of B cell lineages, composing a unique repertoire. Since GCTree generates a small number of productive sequences, we did not use selection models that could reduce the number of generated sequences. Nevertheless, we used a set of parameters to consider the following aspects of the B cell lineage biology: mutability (substitution), tree branching, and base-line mutation rates (Section 2.2.1). To produce simulated repertoires with different SHM loads, we varied the corresponding GCTree parameter $\lambda_0$ that determines the number of mutations in offspring sequences. We experimented with four different configurations {0.16, 0.26, 0.36, 0.46}, where higher values produce more divergent B cell clonal lineages. The simulated benchmarks contain the three types of repertoires: monoclonal, oligoclonal, and polyclonal; Table 1 shows SHM rates, clonality status, the number of sequences, and the number of clonal lineages for all simulated repertoires. Using these simulated benchmarks, we evaluated each MobiLLe parameter configuration by comparing inferred groups to truly related clonal sequences generated during the construction of each simulated repertoire. To evaluate the performance of different parameter configurations, we used the closeness performance measurement, detailed in Section 2.3.1, which evaluated correct group assignment, clonal compositions, and the repertoire structure.

Fig 3 shows closeness F-score distributions of different parameter configurations for each repertoire. Although MobiLLe achieved the best performance (F-score = 1) for all simulated repertoires, we observed more important performance variations on monoclonal and oligoclonal repertoires than polyclonal ones. We also observed an influence of parameter configuration on repertoires produced with higher mutation rates $\lambda_0 = $ {0.36, 0.46}, where we remarked lower F-scores.

To evaluate the influence of parameter configurations on MobiLLe performance, we proceed in three steps. First, we evaluated the impact of the pre-clustering threshold parameter. Then, we checked if the refinement step and 'merge singleton' algorithm could improve MobiLLe performance. Fig 4 shows the overall performance when fixing the pre-clustering $s$ threshold and varying all other parameters. The best parameter configurations were obtained with $s$ in {60%, 70%}, while $s = 50\%$ produced more false positives, see lower values for precision in Fig 4B. On the other hand, higher pre-clustering thresholds produced more false negatives, see lower values for recall in Fig 4C.

Fig 5 shows the impact of the refinement and 'merge singletons' steps on the MobiLLe performance, see Section 2.1.2. To produce these scatter plots, we compared similar parameter configurations that only differ by presence/absence of refinement or 'merge singletons' steps.

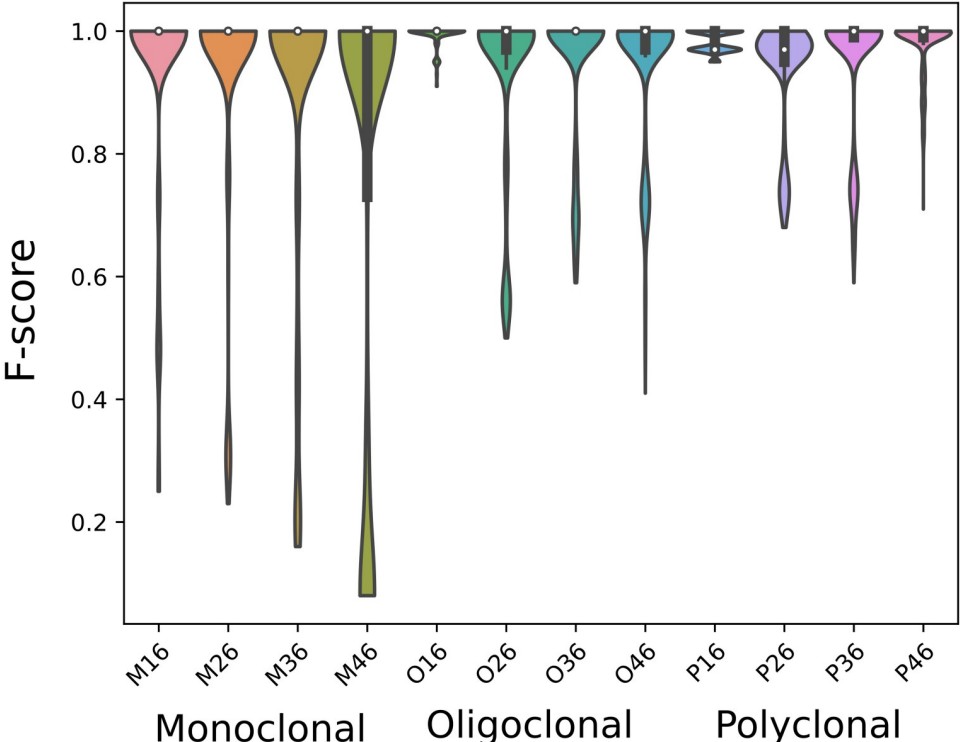

**Fig 3. Performance of different parameter configurations.** We computed the closeness F-score distribution for all simulated repertoire. Each distribution contains 4480 values, one for each parameter configuration. Samples are sorted by repertoire types and SHM rates.

Both plots show the contribution of these two stages, leading to an important improvement in F-scores.

To identify the refinement parameters involved in best/worst MobiLLe performances, we count their frequencies in both situations. We first averaged F-scores of 12 repertoires and ranked them to form two sets of parameter configurations: those with the best performance (highest F-score) and those with the worst performance (lowest F-score). Fig 6 shows the frequency of parameters involved in the two sets. We observed that for IGHV, the binary distance

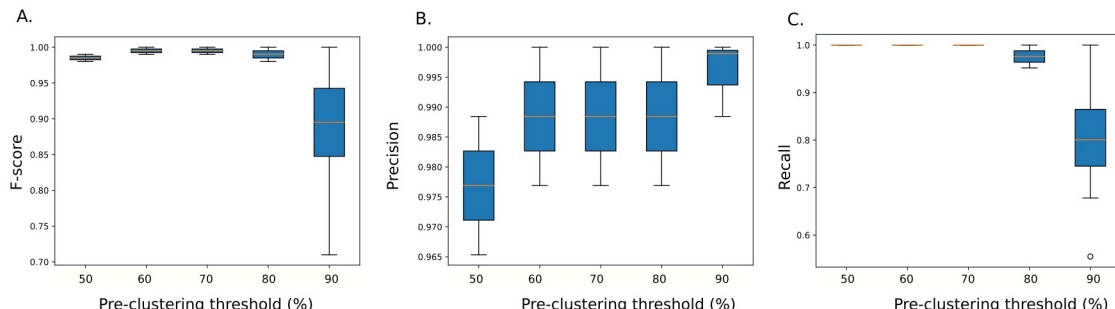

**Fig 4. Effect of pre-clustering threshold on MobiLLe's performance.** The pre-clustering threshold varied from 50% to 90%. We computed the closeness F-score (A), precision (B), and recall (C) distribution by considering all simulated repertoires (53760 parameter configurations).

A.

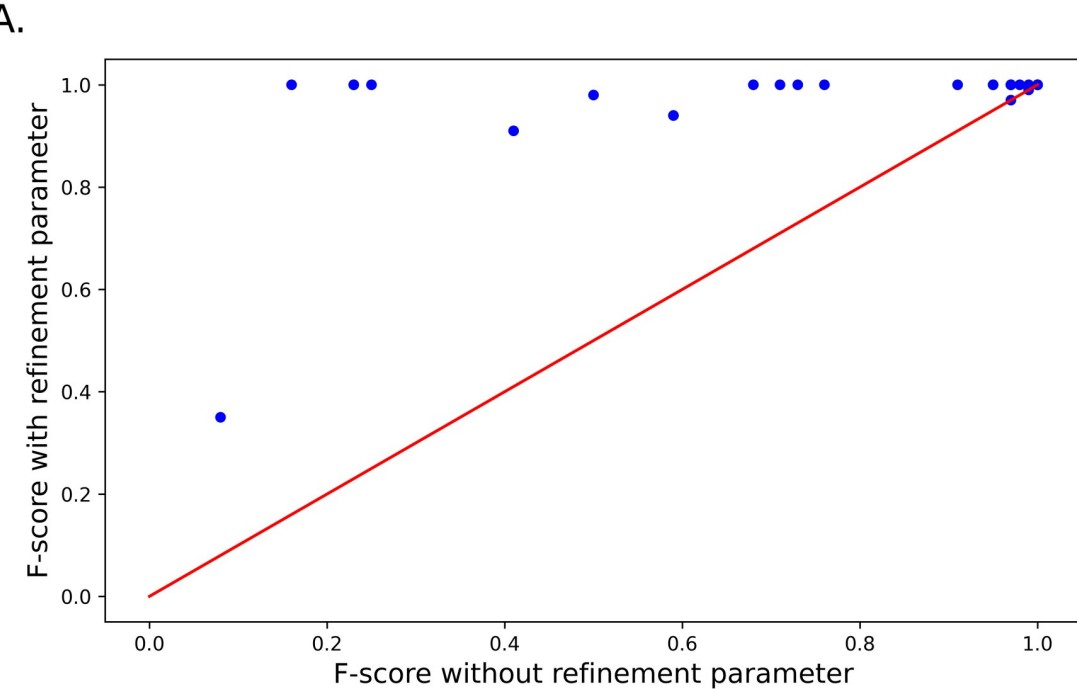

B.

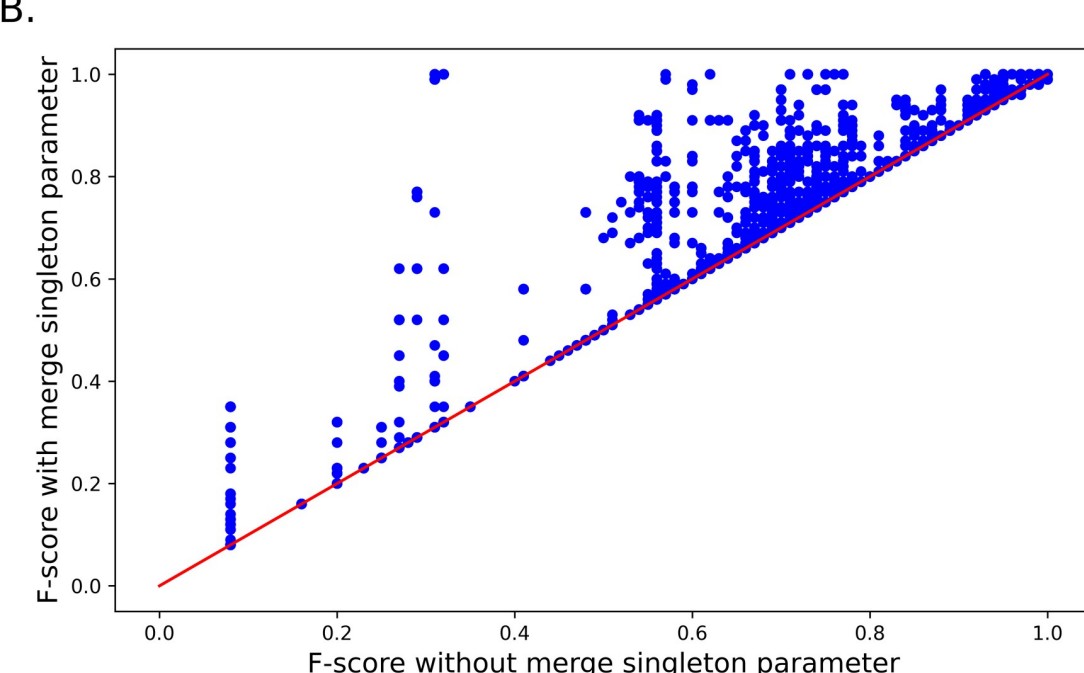

**Fig 5. Importance of using refinement and 'merge singletons' parameters.** A) Scatter plot of MobiLLe F-scores with refinement (ordinate) and without refinement (abscissa) parameter. B) Scatter plot of MobiLLe F-scores with 'merge singletons' (ordinate) and without 'merge singletons' (abscissa) parameter.

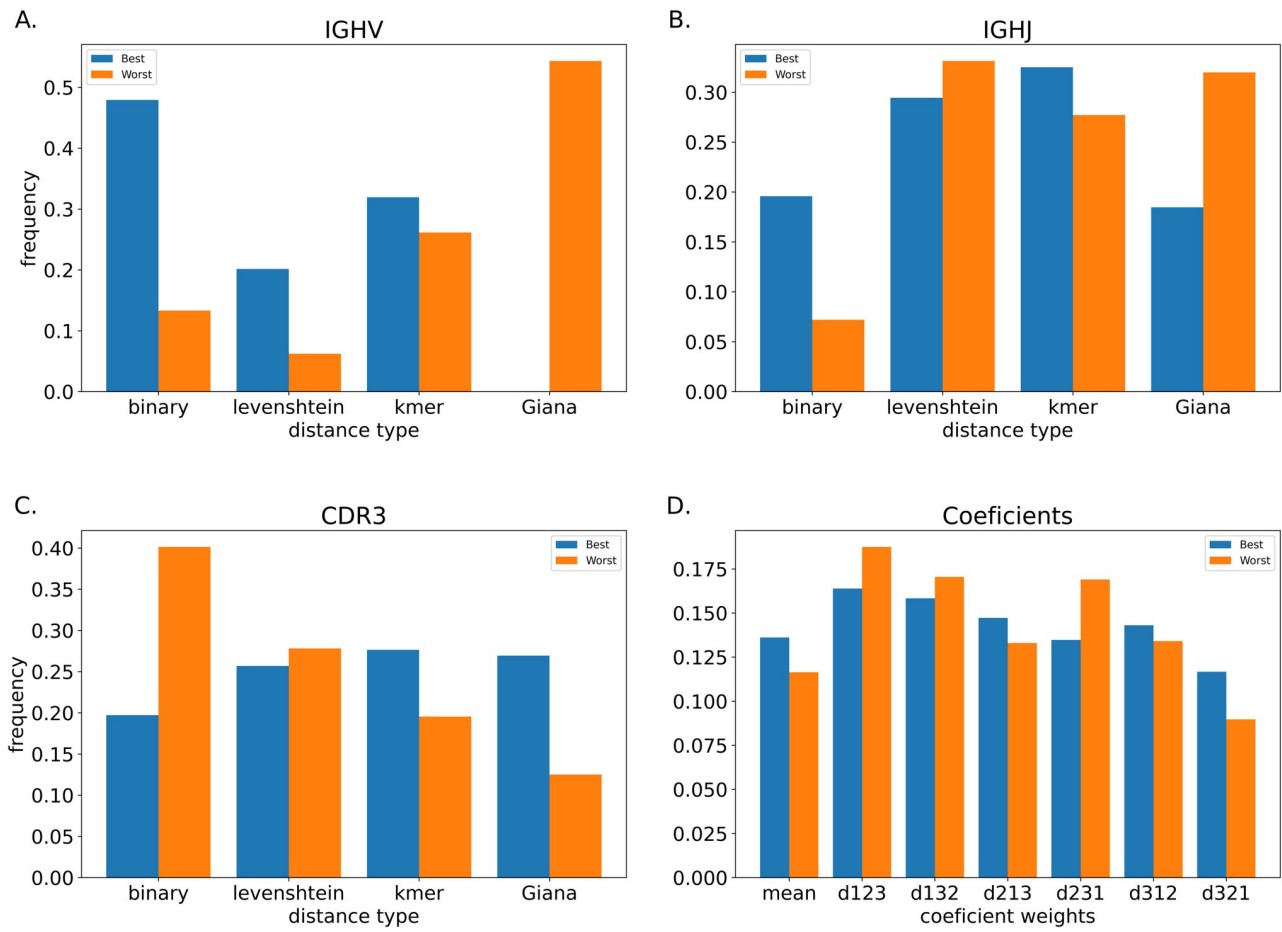

**Fig 6. Impact of refinement parameters in the best and worst performance.** We averaged F-scores of 12 simulated repertoires and ranked them to form two sets of parameter configurations: those with the best performance (F-score = 1) and those with the worst performance (lowest F-score < 0.7). The ordinate shows parameter frequency and abscissa parameter type. (A, B, and C) show IGHV, IGHJ, and CDR3 distances, while (D) shows coefficient variations. Note that d––– indicates the coefficient values for $\alpha$, $\beta$, and $\lambda$ respectively, while 'mean' represents the arithmetic mean.

achieved the best performance, being more present in the 'best' set than the 'worst' set, Levenshtein and k-mer-based distances were also more frequent in the 'best' set, while GIANA was just involved in 'worst' set. For IGHJ, the k-mer based distance produced the best results, but we also observed its participation in the 'worst' set. Binary distance seems more beneficial than disadvantageous, while we observed the opposite with Levenshtein and GIANA distances. Binary distance produced the worst results for CDR3, while GIANA distance produced more 'best' than 'worst' results. For the coefficients, we observed that the arithmetic mean produced satisfactory results, and higher $\alpha$ produced more 'best' than 'worst' results.

Generally, the method was very robust; on average, 77% of parameter configurations achieved F-score $\geq$0.98. Thus, we choose an equally performing and time-efficient configuration as default parameters to further apply it to experimental datasets. For the rest of this work, we used $s$ = 70%, a binary distance for IGHV, Levenshtein for IGHJ and CDR3 distances, and arithmetic means for combining later distances. Finally, we ran MobiLLe with refinement and the 'merge singletons' options.

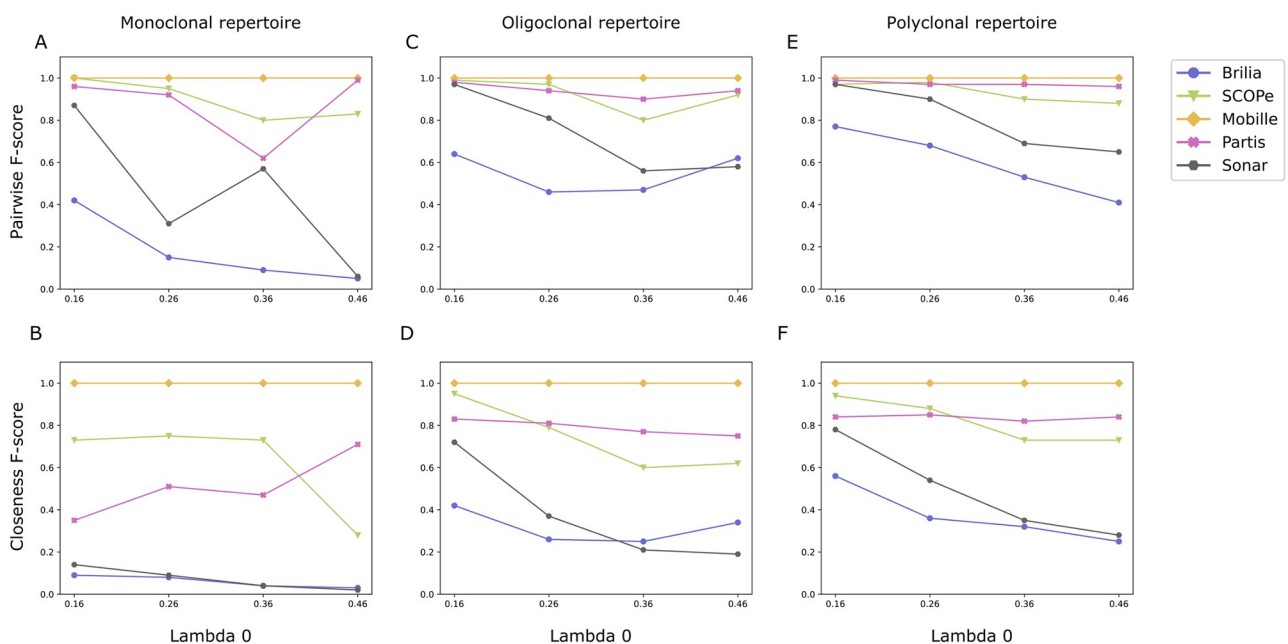

**Fig 7. Comparison of clustering accuracy on simulated repertoires.** Performance evaluation of five different BCR lineage grouping methods on 12 simulated repertoires.

## 3.2 Comparison to existing BCR lineage grouping methods

To obtain comparative results of different clonal lineage grouping methods, we compared the performance of MobiLLe and four state-of-art methods on simulated, artificial, and experimental data. We considered Partis, SCOPe, SONAR, and BRILIA, briefly described in Section 2.4, and used the evaluation strategies described in Section 2.3 to measure their performances and compare their clonal distributions.

**3.2.1 Simulated BCR repertoires.**   We first compared MobiLLe to the four selected methods on the 12 simulated repertoires described in Section 2.2.1. To quantify clustering accuracy, we used precision, recall and F-score computed in two ways: pairwise and closeness (Section 2.3.1).

**Pairwise performances.**   MobiLLe achieved the best pairwise performance across all simulated datasets, Fig 7 Top. SCOPe and Partis outperformed SONAR and BRILIA. All tools achieved a precision close to 1, with few false positives. However, most tools over-split clonal lineages, detecting many false negatives that considerably decreased their recall and F-score values.

SCOPe achieved high recalls and F-scores for simulated benchmarks with lower mutation rates ($\lambda_0 = \{0.16, 0.26\}$), see S2–S7 Tables. Recall and F-score values were above 0.94 for these six simulated repertoires. We observed lower recalls and F-scores for the remaining benchmarks, produced with higher mutation rates $\lambda_0 = \{0.36, 0.46\}$ (see S8–S13 Tables). On the other hand, Partis obtained a good pairwise performance across all simulated benchmarks independently of the mutation rate. The only exception was the monoclonal repertoire M36 (S8 Table). Partis detected 44 groups for this benchmark when the expected was 35, decreasing its recall considerably. Interestingly, for lower mutation rates, SCOPe outperformed Partis, but we observed the reverse for higher mutation rates for most simulated repertoires. Thus, Partis seems more accurate when analyzing clonally-related sequences with higher divergence. It had

difficulties separating clusters with highly similar naive sequences, as reported in the original publication.

The different mutation rates impacted SONAR performances for oligoclonal and polyclonal repertoires. Recalls and F-scores decreased as long as mutation rates increased, especially for the oligoclonal repertoires. For the monoclonal samples with $\lambda_0 = 0.26$ (S5 Table), SONAR obtained lower recall and F-score than with $\lambda_0 = 0.36$ (S8 Table). We observed that SONAR over-split the largest clonal group of the first repertoire ($\lambda_0 = 0.26$), grouping only 37% of sequences. On the other hand, it splits less the most abundant clonal lineage of monoclonal repertoire generated with $\lambda_0 = 0.36$, grouping 62% of sequences. As splits in large clonal lineages contribute more to accuracy decreasing, it could explain the lower performance of SONAR on the monoclonal repertoire ($\lambda_0 = 0.26$). For the monoclonal repertoire with $\lambda_0 = 0.46$, SONAR detected four times more clonal groups than expected, obtaining its lowest recall and F-score, 0.03 and 0.06, respectively.

Most of the time, BRILIA achieved the lowest pairwise performances across all simulated repertoires generated with different mutation rates. BRILIA removes sequences that it cannot annotate, thus reducing the original dataset, which impacts the accuracy calculation. We also observed that BRILIA over-split repertoires, producing the highest number of clonal lineages for most simulated benchmarks. The best performance was obtained on polyclonal repertoires generated with lower mutation rates ($\lambda_0 = \{0.16, 0.26\}$, S4–S7 Tables) and the lowest performance on monoclonal repertoires with higher mutation rates ($\lambda_0 = \{0.36, 0.46\}$, S8 and S11 Tables).

**Closeness performances.** MobiLLe achieved the best closeness performance across all simulated benchmarks independently of mutation rates; see Fig 7. The closeness evaluation approach is more challenging than the pairwise approach since clonal lineage properties such as size and distribution are also evaluated rather than pairwise relationships. Conversely, MobiLLe performance was not impacted by higher mutation rates or unbalanced clonal distributions, as observed for other tools. MobiLLe reconstructed all repertoire structures precisely, showing good stability and high accuracy. On the other hand, the four evaluated BCR lineage grouping tools obtained high precision values but much lower recalls and F-scores. As observed for pairwise measures, SCOPe and Partis outperformed SONAR and BRILIA.

SCOPe outperformed Partis for most monoclonal repertoires, but Partis surpassed SCOPe for the majority of oligoclonal and polyclonal samples. SCOPe achieved higher F-score values ($> 0.73$) on monoclonal repertoires generated with lower mutation rates ($\lambda_0 = \{0.16, 0.26, 0.36\}$, see S2, S5 and S8 Tables). However, its performance sharply decreased on the monoclonal repertoire with the highest mutation rate, achieving 0.16 and 0.28 for recall and F-score (S11 Table). This was particularly the case on oligoclonal and polyclonal samples, where we observed a significant difference between repertoires generated with $\lambda_0 = \{0.16, 0.26\}$ and those generated with $\lambda_0 = \{0.36, 0.46\}$; Fig 7D and 7F. Higher mutation rates did not impact the performance of Partis on simulated repertoires. Its accuracy was stable on polyclonal repertoires and presented slight fluctuations on oligoclonal samples. Interesting, on the monoclonal benchmarks, Partis achieved better performance for highly mutated repertoires, achieving its best F-score on the sample generated with $\lambda_0 = \{0.46\}$; Fig 7B and S11 Table.

SONAR performance was greatly affected by high mutation rates. We systematically observed lower recalls/F-scores as mutation rates increased (Fig 7D and 7F and S2–S13 Tables). Independently of mutation rates, SONAR achieved low performance on monoclonal repertoires; F-scores were smaller than 0.2 (S2, S5, S8 and S11 Tables). SONAR over-split the most abundant clonal group of monoclonal repertoires that greatly decreased closeness performances. We observed better results on oligoclonal and polyclonal repertoires, especially on

samples generated with lower mutation rates ($\lambda_0$ = {0.16, 0.26} (S3 and S6 Tables). On the other hand, for all repertoires generated with higher mutation rates, SONAR achieved an F-score inferior to 0.4 (S8–S13 Tables). BRILIA achieved the lowest performance for most of the analyzed repertoires. The only exception was the oligoclonal sample generated with $\lambda_0$ = 0.46, where it outperformed SONAR (S12 Table). For the remaining benchmarks, SONAR over-passed BRILIA; we observed a notable difference mainly on repertoires generated with lower mutation rates, where BRILIA achieved lower F-scores systematically (S3, S4, S6 and S7 Tables). BRILIA and SONAR achieved an equivalent performance in monoclonal repertoires, with shallow values for recall and F-score.

**3.2.2 Artificial experimental repertoires.**   To investigate the tools' performance on experimental benchmarks, we created artificial repertoires from BCR high throughput sequencing data as described in Section 2.2.2. We generated three artificial monoclonal repertoires (AMR1, AMR2, and AMR3) by mixing sequences from a pure B cell lineage (10%) and a polyclonal background (90%). Each benchmark contains 10k sequences, and the performance measures clonal grouping tools' ability to identify memberships within the most abundant group (the pure B cell lineage). Accurate tools might group sequences from the most abundant group and separate those from the polyclonal background in different clusters. Thus, we counted the number of split clusters (SC) and false positives (FP) of the most abundant group to measure the tools' accuracy. We also used the alluvial diagrams for visualizing clustering results; it represents flows between expected clonal lineages (left) and inferred ones (right).

Fig 8 shows the performance of different tools for AMR1, AMR2, and AMR3. In each alluvial diagram, blue blocks (on the left) represent the pure B cell lineage, and pink or orange (on the right) inferred clonal lineages. Pink blocks contain only sequences of the pure B cell lineage (true positives), while the orange blocks sequences from the polyclonal background (false positives). Block height symbolizes the size of a clonal lineage, that is, the number of sequences. For AMR1, MobiLLe obtained the best clonal separation with just one split (SC = 1) and no false positive (FP = 0). Partis and SCOPe obtained few false positives but higher SC, 4 and 5, respectively. SONAR and BRILIA did not yield any false positives, but both tools produced a significant number of splits, 90 and 64, respectively. Note that the only sequence that MobiLLe left out had a different CDR3 length and less than 70% of amino acid sequence identity. All other tools also placed this sequence outside of the major clonal lineage.

MobiLLe did not split the AMR2 B cell lineage sequences into different groups (SC = 0). It also obtained the lowest FP value compared to Partis and SCOPe, which similarly did not divide the most abundant clonal lineage. SONAR and BRILIA still over-split the pure B cell lineage sequences into several groups, achieving an SC of 90 and 55, respectively. SONAR detected more groups than BRILIA but obtained fewer FPs. Notably, the pure B cell lineage of AMR3 contains non-productive sequences that SONAR and BRILIA did not consider. MobiLLe recovered all sequences of the most abundant clonal lineage (SC = 0), while SCOPe and Partis performed 2 and 3 splits, respectively, see Fig 8C. For all artificial monoclonal repertoires, we observed the same behavior in the results of each tool. We noted that BCR lineage grouping methods clustered sequences differently. Partis, SCOPe, and MobiLLe grouped most sequences from pure B cell lineages. SONAR and BRILIA over-split clonally-related sequences but detected fewer FPs than MobiLLe on benchmarks AMR2, and fewer FPs than Partis and SCOPe on all benchmarks. MobiLLe presented the best performance, achieving minimum splits and false positives.

**3.2.3 Clonal distribution comparisons.**   In order to understand the differences in clonal distributions of experimental repertoires, we compared MobiLLe cluster composition to each considered BCR lineage grouping tool. For comparing the methods, we selected three repertoires with different clonality status (Section 2.2.3 and S5 Fig). $I_1$ is a monoclonal repertoire

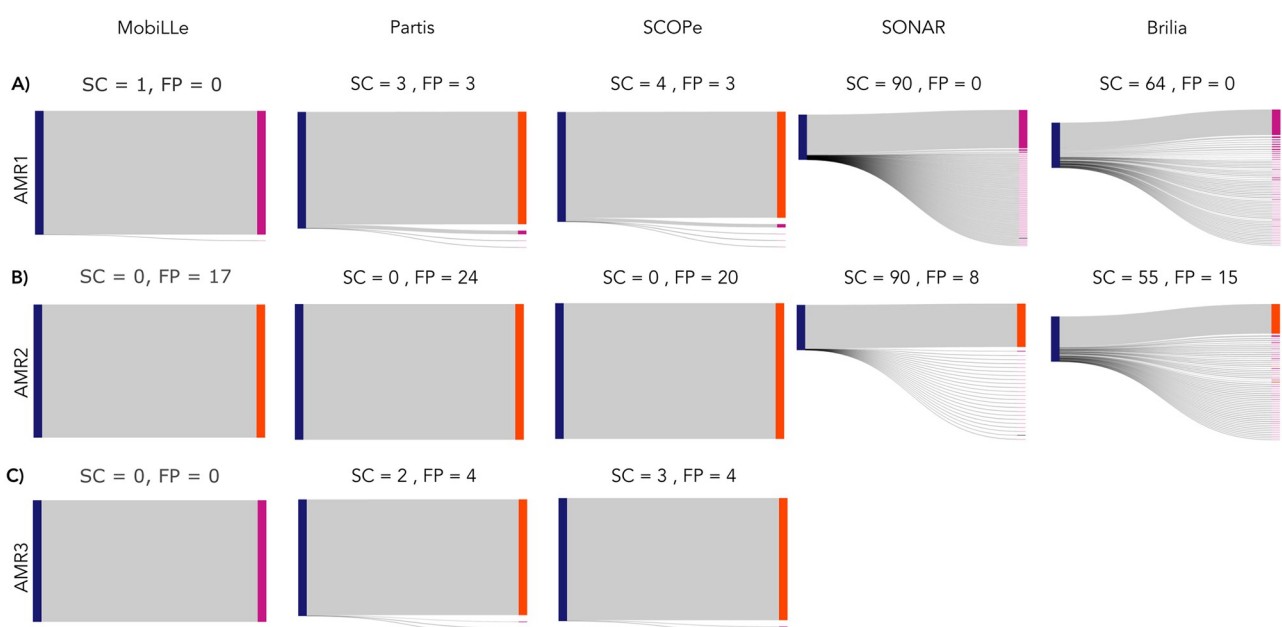

**Fig 8. Performance comparison on artificial monoclonal repertoires.** We generated three artificial monoclonal repertoires (AMR1, AMR2, and AMR3) by sampling sequences from a pure B cell lineage (10%) and a polyclonal background (90%). Each benchmark contained 10000 sequences. Accurate tools might group sequences from the pure B cell lineage and separate those from the polyclonal background. We measured the performance of BCR lineage grouping methods by computing the number of splits (SC) and false positives (FP) of the most abundant group. To better visualize and compare clustering results, we show alluvial diagrams for AMR1 (a), AMR2 (b), and AMR3 (c), where blue blocks represent the pure B cell lineage and pink or orange inferred groups. Pink blocks contain only sequences belonging to the pure B cell lineage (true positives), while the orange blocks contain sequences from the polyclonal background (false positives). SONAR and BRILIA did not produce results for the AMR3 benchmarks since they do not deal with non-productive sequences.

where the most abundant clonal group contains 98% of all sequences, $I_6$ is oligoclonal, and $I_8$ is a polyclonal repertoire; see Table 2 and S5(A), S5(F) and S5(H) Fig. We compared the inferred clonal lineages of existing grouping tools with MobiLLe's clustering results. For that, we defined five events: identical, join, split, mix, and "not found," which represent the (dis) similarities between two clonal distributions, see Section 2.3.2 and Fig 2. Fig 9 shows the occurrences of such events when comparing MobiLLe against Partis, SCOPe, SONAR, and BRILIA. Note that we also quantified the number of MobiLLe's missing clonal lineages as a "Not found" event since MobiLLe removes sequences with no V(D)J annotations. To better understand event results, we also computed pairwise performance between MobiLLe and each evaluated tool (Section 2.3.1) by comparing the inferred groups—obtained by other tools—to the MobiLLe groups.

When analyzing the repertoire $I_1$, we observed that BRILIA presented the maximum number of identical clusters, followed by Partis and SCOPe. These three tools obtained a similar number of clusters in the order of magnitude of the MobiLLe output. On the other hand, SONAR presented a significantly higher number of clusters, performing many splits. Interestingly, BRILIA and Partis achieved very high pairwise performances (S14 Table), indicating that most pairs of clonally-related sequences were identically clustered. SCOPe inferred more clonal lineages than MobiLLe, achieving a slightly lower pairwise performance but around 0.98 (S14 Table). As expected, the recalls and F-scores of SONAR were very low since it performed a considerable number of splits. All compared tools produced fewer joins, indicating that they did not group sequences separated by MobiLLe.

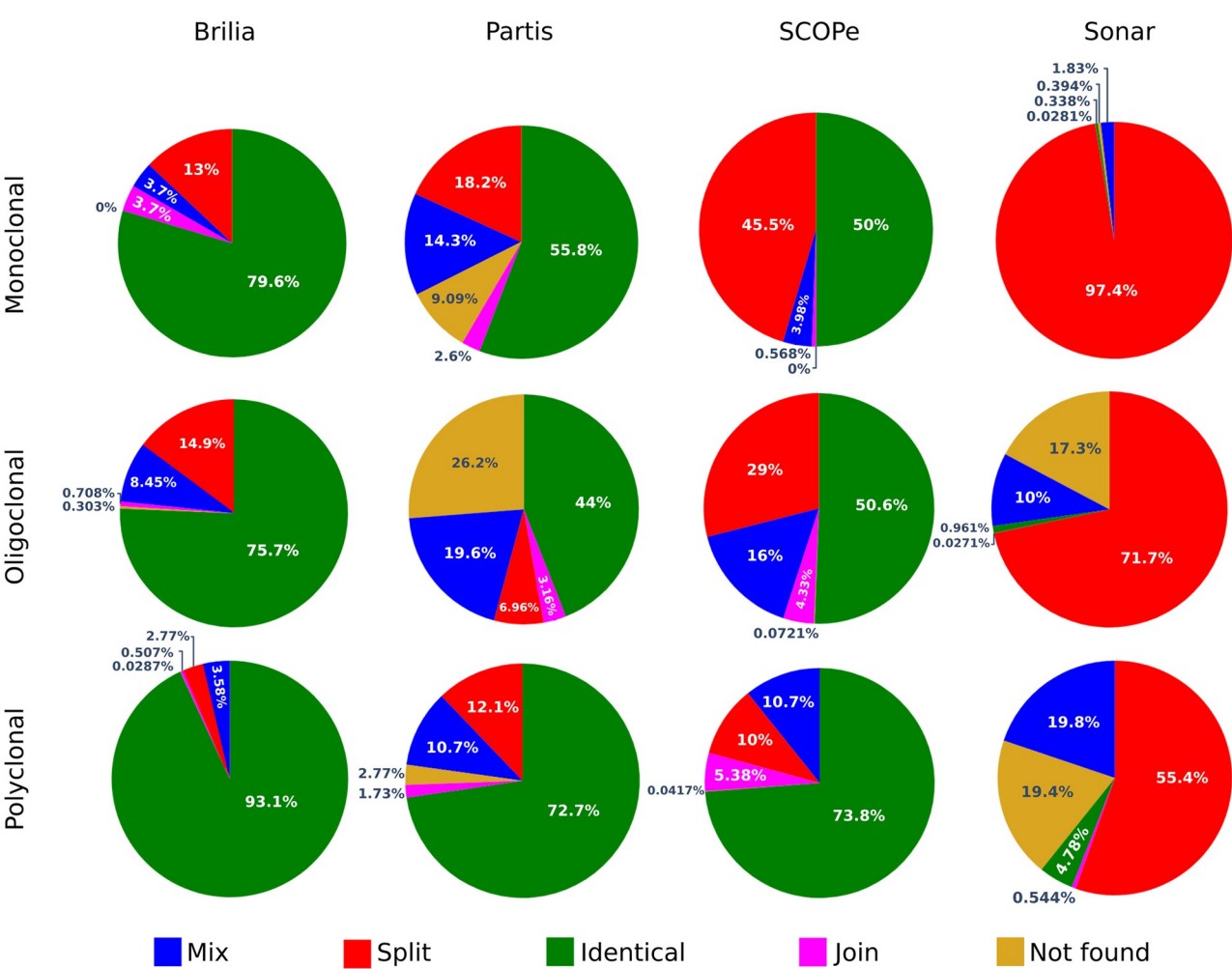

**Fig 9. Clonal distribution comparisons on three experimental repertoires.** We compared the inferred clonal lineages of each BCR lineage grouping tool with MobiLLe's clustering results. For that, we defined five events: identical, join, split, mix, and not found, representing the (dis)similarities between two clonal distributions: $d_1$ (MobiLLe) and $d_2$ (another tool). The "identical" event accounts for the percentage of identical clonal lineages found in both distributions; the "join" event reports the percentage of $d_1$ clonal lineages found merged in $d_2$ while "split" the percentage of $d_1$ clonal lineages found separated in $d_2$. The "mix" event accounts for a mixture of "join" and "split" events while "not found" reports the percentage of clonal lineages in $d_2$ not found in $d_1$; see an illustration in Fig 2.

For the oligoclonal repertoire $I_6$, BRILIA inferred the highest number of identical clonal lineages, approximating the number of clonal lineages predicted by MobiLLe. For Partis and SCOPe, the predominant event was also "identical," but both tools inferred fewer clonal lineages than MobiLLe. For SCOPe, we also observed that the "mix" event was frequent, accounting for 29% of clonal groups. Further analyses showed that "mix" events were associated with minor differences involving singletons. It was confirmed by pairwise performance values (S14 Table) showing very high F-scores for SCOPe and Partis. SCOPe generated more splits than BRILIA and Partis, but it did not impact its pairwise measures since most splits were not in the most abundant clonal group as observed for SONAR. SONAR over-split, detecting three times more clonal lineages than MobiLLe. Consequently, its pairwise recall 0.18 was very low (S14 Table). Its precision was also very low, showing many differences compared to MobiLLe.

## Clustering computational time

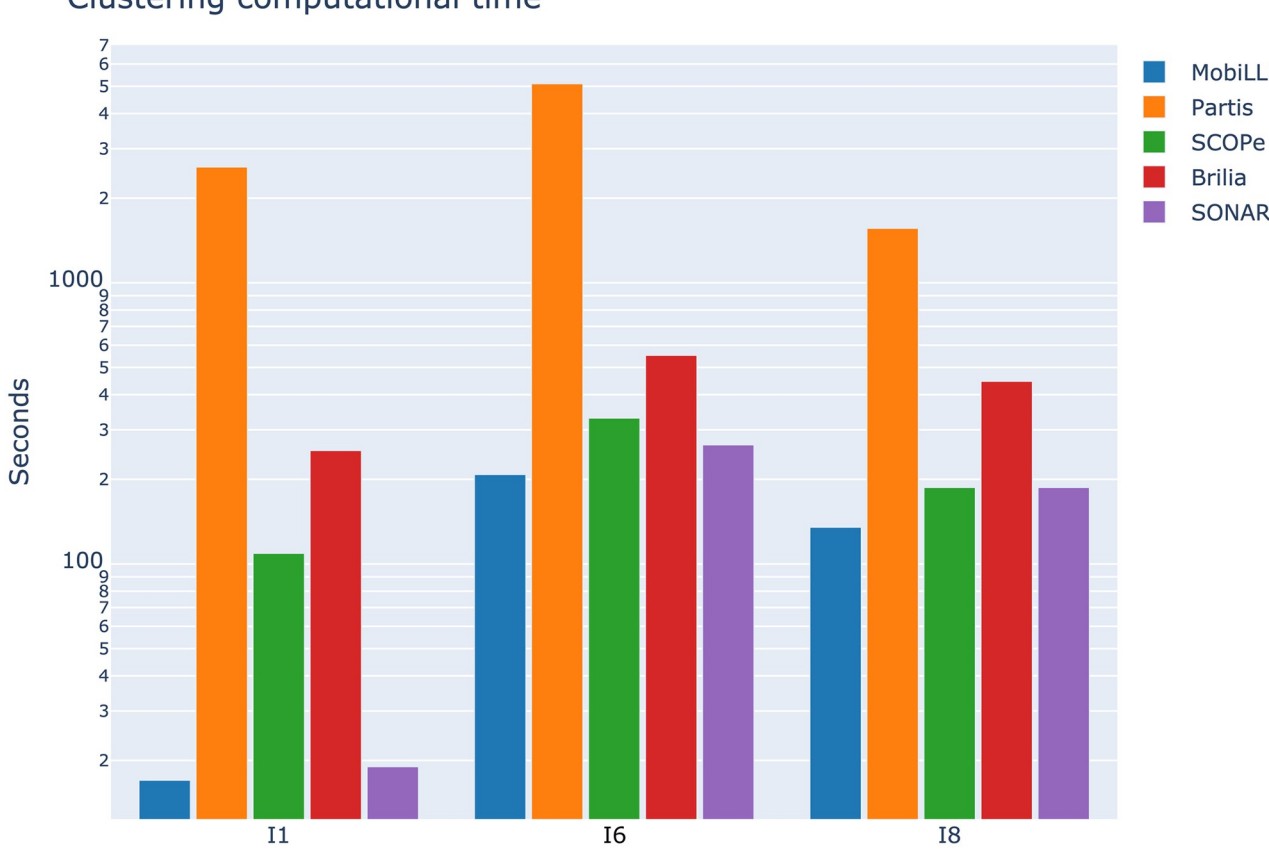

**Fig 10. Comparing running times of clonal grouping tools.** The running times for MobiLLe and other tools were measured for three experimental repertoires with different clonal compositions. To a better visualisation, we used log scale, S16 Table shows the time in second for each considered tool.

We observed more identical inferred clonal lineages in the polyclonal repertoire $I_8$ for BRILIA, Partis and SCOPe. BRILIA presented the highest value, accounting for more than 93%, followed by SCOPe and Partis. BRILIA and Partis obtained the best pairwise values. There was a good agreement between Partis and MobiLLe, but the former produced more splits than BRILIA and SCOPe, resulting in a slightly decreased pairwise recall. In general, the clonal composition of the three tools was similar to MobiLLe (S15 Table). On the other hand, SONAR over-split the clonal lineages inferred by MobiLLe, detecting the largest number of groups. The clonal composition was also very different, resulting in low precision, (S14 Table).

**3.2.4 Runtime.** Computational efficiency is an important factor in BCR lineage grouping. Efficient methods need to process a large number of BCR sequences within a reasonable time without compromising clustering quality. In order to compare the computational requirements of MobiLLe with the four selected BCR lineage grouping tools, we measured the running time on the $I_1$, $I_6$ and $I_8$ repertoires. Fig 10 shows the time in seconds required by each tool to process the three repertoires, using a 3.4 GHz Octa-Core processor with 32 GB of memory; see the exact time in S16 Table. MobiLLe took less than 20s to analyse the monoclonal repertoire $I_1$ that contains more than 30000 sequences. SONAR was also quick, but we observed an over-split that could explain its faster performance. BRILIA and SCOPe analysed the $I_1$ repertoire in a comparable time, around 200s and 100s, respectively. Partis was the most time-consuming tool, taking more than 2000 seconds to process the $I_1$ repertoire.

For $I_6$, an oligoclonal repertoire containing more than 140K sequences, MobiLLe also exhibited the fastest performance, but this repertoire required the longest processing time. Although $I_6$ presents an oligoclonal structure, it has a clonal lineage that groups more than 14% of sequences. The $I_6$ repertoire structure slowed down the clustering task since more comparisons among higher density clusters were required. Note that all evaluated tools spent more time analysing the $I_6$ repertoire. The polyclonal repertoire $I_8$, containing more than 70000, appeared slightly less challenging than $I_6$ with fewer sequences. Again, MobiLLe achieved the best time performance, clustering $I_8$ in 135s. SONAR and SCOPe were also fast, taking 187s. Partis achieved the lowest performance, taking more than 1500s.

We found that the clonal distribution significantly influences the running time of MobiLLe rather than the number of sequences. Calculating intra-clonal distances in repertoires with high abundant clonal lineages was more time-consuming. Accordingly, for repertoires with similar sizes but different clonal distributions, MobiLLe can present different running times.

## 3.3 Exploring clonality in the repertoire of normal individuals and patients with hematological malignancies

A variety of clinical assays are available to detect the presence of B cell clonal expansion (clonality), helping to diagnose, for instance, lymphomas and leukemias. Although these strategies are adequate for many applications, they do not explore essential features inherent in rearranged immune receptor gene sequences. Several studies using deep sequencing of BCR genes revealed intraclonal heterogeneities in a subset of cases, while priors approaches could not detect them [39, 40]. To check if MobiLLe can accurately detect clonality status in patients with lymphoid malignancies, we tested it on nine experimental repertoires from the Pitié-Salpêtrière hospital (Paris-France), and five public repertoires available at IReceptor repository [32].

**3.3.1 Comparing standard clonality clinical assays with MobiLLe.** In order to compare MobiLLe results to standard clonality assays, we selected nine samples of human peripheral blood mononuclear cells collected during routine diagnostic procedures at the Pitié-Salpêtrière hospital. Three were leukemic samples (chronic lymphocytic leukemia, CLL), and six of them were considered non-clonal, originating from patients devoid of malignancy (Section 2.2.3). Their clonality status had been previously established by conventional methods, including PCR amplification of IGH-VDJ rearrangements followed by Genescan analysis [33] (S5 Fig).

Fig 11 shows the clonal distribution for each analyzed sample; see also S7 Fig. To measure the disequilibrium of a repertoire, we computed the Gini coefficient [41], which reflects the inequalities among values of a frequency distribution; zero indicates perfect equality, while one corresponds to maximal inequality, see panel C of S8–S16 Figs. Clonal repertoires presented the highest Gini indices, close to 1 for individuals 1 to 3 (compare panels C of S8–S10 Figs). Repertoires 1 and 3 presented similar clonal distributions, with the presence of a major clonal group representing the quasi-totality of the repertoire and a small number of minor clonal lineages having a low number of sequences (compare I1 and I3 in Fig 11 and S8 and S10 Figs). MobiLLe results were concordant with Genescan analysis, where a single pick was observed, indicating a monoclonal repertoire profile (S5(A) and S5(C) Fig).

Individual 2 presented a different clonal distribution with two major clonal lineages, each one accounting for more than 40% of the repertoire, see I2 in Fig 11 and S9 Fig). Detailed sequence analysis revealed that the two major clonal lineages were composed of a productive and an unproductive IGH-VDJ rearrangement, corresponding to a leukemic cell population with biallelic IGH rearrangements. This was also evident in Genescan analysis; S5(B) Fig, two peaks are observed, each corresponding to one IGH allele.

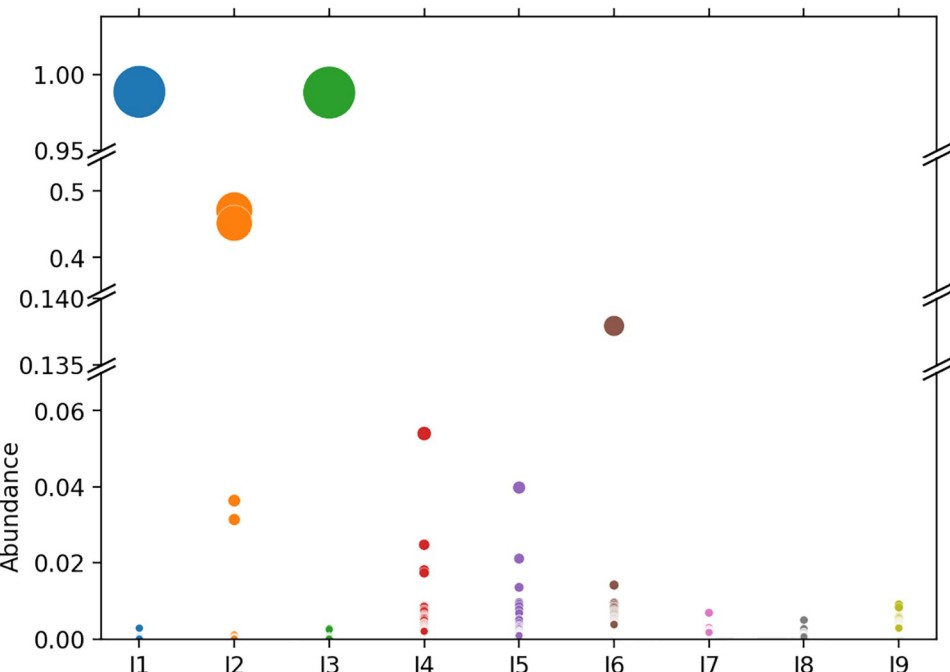

**Fig 11. Clonal distribution/density of nine experimental repertoires.** We plotted the 20 most abundant clonal lineages for each repertoire; Circles represent clonal groups, while their areas are proportional to the clonal group abundance.

To better analyze non-clonal cases, we split them into two groups: (i) those with the predominance of several clonal lineages of moderate abundance: $I_4$, $I_5$ and $I_6$ (considered as minor clonal or oligoclonal repertoires), and (ii) the others with a more equilibrated (balanced) sequence distribution: $I_7$, $I_8$ and $I_9$, representing polyclonal repertoires. In the first group, repertoires 4 and 5 displayed similar clonal distributions, with Gini indices around 0.76 and 0.84, respectively, see Fig 11, S11 and S12 Figs. GeneScan analyses showed only minor peaks emerging above the polyclonal background, S5(D) and S5(E) Fig. Individual 6 presented a slightly different configuration from other repertoires in the same group, with the presence of a clonal lineage representing around 13% of the repertoire, Fig 11 and S13 Fig. Unsurprisingly, the repertoire $I_6$ displayed the most biased distribution among non-clonal repertoires with a Gini index of 0.91. We also observed a more irregular profile in the Genescan analysis, (S5(F) Fig).

In the second group, we found more homogeneous and less biased repertoires. Sample 7 and 8 had similar clonal distributions (Fig 11 and S14 and S15 Figs), while repertoire 9 was more irregular (Gini index around 0.83), see Fig 11 and S16 Fig. We also observed a slight difference when comparing distributions generated by Genescan analysis, where 9 displayed a more irregular polyclonal profile, compare S5(G)–S5(I) Fig. In these three cases, the size of the detected clonal lineages was small, each of them accounting for less than 1% of the total sequences; see S14–S16 Figs.

**3.3.2 Characterizing B cell populations from different lymphoproliferative diseases.** To further evaluate MobiLLe, we selected an independent series of lymphoproliferative disorder repertoires from a previously published study [34] with available sequence data. Samples were collected from distinct tissues, see Table 3. Five cases were selected, including one patient diagnosed with follicular lymphoma/small lymphocytic lymphoma (P3-FL-SLL), one with

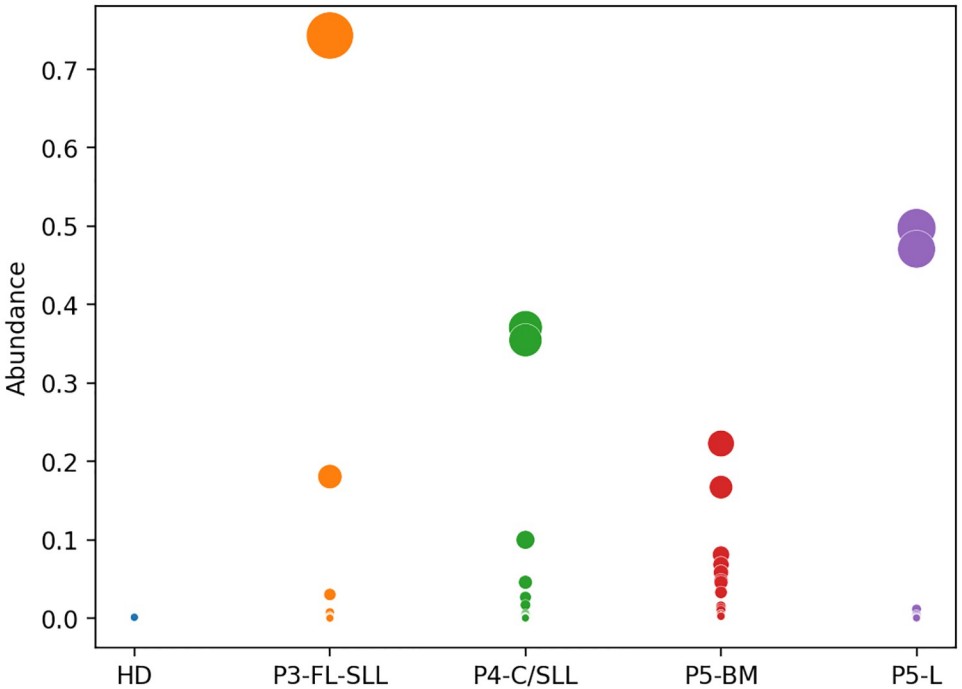

**Fig 12. Clonal distributions of a healthy donor and individuals with different lymphoproliferative diseases.** We plotted the 20 most abundant clonal lineages for each repertoire; Circles represent clonal groups, while their areas are proportional to the clonal group abundance. Report to Table 3 for repertoire properties and individuals' labels.

chronic lymphocytic leukemia/small lymphocytic lymphoma (P4-C/SLL), and two samples from the same patient diagnosed with a post-transplant lymphoproliferative disorder: P5-L was sampled from the liver, while P5-BM from bone marrow. The fifth sample was from a healthy donor (HD).

Fig 12 compares the clonal distribution of each repertoire's 20 most abundant clonal lineages; see also S17 Fig. As expected, we did not observe any clonal expansions in the HD repertoire. On the other hand, we observed a considerable clonal expansion in the repertoires of all patients. For P3-FL-SLL, MobiLLe detected two distinct V(D)J rearrangements; the most abundant clonal lineage grouped 74% of IGH sequences and used gene segments: IGHV1–18*01 and IGHJ4*02, while the second clonal B cell population grouped 18% of sequences and used IGHV3–30*02 and IGHJ4*02 genes. MobiLLe clonal distributions for these five repertoires agreed with the original paper [34]; their results were supported by morphological and immunophenotypic evidence, showing two different B cell lymphomas (follicular lymphoma and small lymphocytic lymphoma) in the tissue.

The analysis of the P4-C/SLL repertoire showed an oligoclonal population with several clonal lineages with an abundance superior to 1%. These results agreed with the original publication, where they observed several groups represented by large points. The two most abundant clonal lineages grouped 37% and 35% of IGH sequences and used very different segment genes, IGHV4–61*02/IGHJ6*03 and IGHV1-NL1*01/IGHJ5*02, respectively. The other five most abundant clonal lineages used different genes with very different CDR3 regions.

Patient 5 was a challenging case, as reported in the original publication. The patient had undergone a liver transplant and later developed a large B cell lymphoma in the liver, a manifestation of a post-transplant lymphoproliferative disorder, a condition in which

immunosuppression leads to B cell lymphomas, usually associated with Epstein-Barr virus infection. MobiLLe analyzed the two samples of patient 5 collected from bone marrow (P5-BM) and the liver (P5-L). We observed clonal expansions in both, being more abundant in P5-L than P5-BM, as shown in Fig 12 and S17(D) and S7(E) Fig. The original publication showed that capillary electrophoresis sizing of V(D)J rearrangements confirmed the presence of a clonal population in both samples. However, only sequencing data uncovered no relationship between the clonal lineages found in the liver and those in the bone marrow B cells. The most abundant clonal lineage of P5-BM grouped 22% of the repertoire sequences and used: IGHV3–15*01 and IGHJ6*02 genes, while P5-L grouped 50% of IGH sequences and used IGHV1–69*06 and IGHJ4*02 genes. Of note, it is not unusual to observe independent EBV-driven clonal B-cell proliferations in the context of post-transplantation immune suppression.

## 3.4 MobiLLe identifies clonal expansion of B cells in severe/moderate COVID-19

Previous work has suggested that clonal B cell expansions can be detected in response to the SARS-CoV-2 infection [35]. We sought to determine whether MobiLLe could detect such clonal expansions in a non-malignant context. To address this issue, we selected 12 repertoires from that study and downloaded their IGH sequences from IReceptor [32]. The 12 repertoires correspond to three healthy donors, three patients with moderate COVID-19, and six severe COVID-19, see details in Section 2.2.3 and Table 4. Next, we ran MobiLLe to identify clonal distribution for these repertoires and evaluated their clonal expansions.

Fig 13 compares the clonal distribution of the 20 most abundant clonal lineages of 12 repertoires, see also S18 and S19 Figs. We observed in Fig 13A a more important clonal expansion for patients with severe and moderate COVID-19 than healthy donors. This was concordant with the original publication results, thus confirming the ability of MobiLLe to identify such clonal expansions in patients with severe and moderate COVID-19, and distinguish them from healthy donors. Fig 13B shows the clonal expansion of each individual. Repertoires were ranked according to the sum of their most abundant clonal lineages. We obtained the same

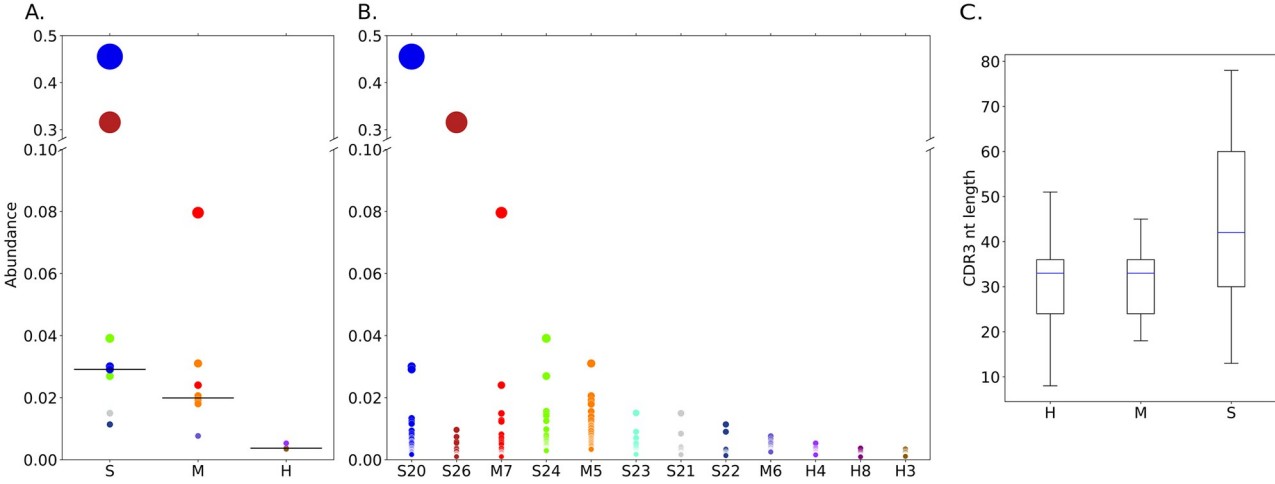

**Fig 13. Clonal distribution of healthy donors and patients with moderate/severe COVID-19.** A) Abundance of top 20 ranked clonal groups stratified by clinical status. We plotted each individual's most abundant clonal groups until achieving 20 samples. B) Abundance of top 20 ranked clonal groups stratified by individuals. C) CDR3 nucleotide lengths of the top 20 clonal groups, stratified by clinical status. In panels A and B, circles represent clonal lineages, while their areas are proportional to the clonal group abundance. Each color represents an individual. Report to Table 4 for repertoire properties and individuals' labels. Abbreviations: S-severe and M-moderate COVID-19; H-healthy donors.

order of the original publication, except for patient S23, who appeared before M5. We also confirmed that in individuals with severe disease, CDR3 sequences exhibited greater variation in length, Fig 13C. We demonstrated that MobiLLe could analyze non-malignant repertoires such as COVID-19 data and help associate severe COVID-19 with large and oligoclonal B cell expansions, having divergent CDR3 sequences.

## 4 Discussion

The ability to obtain a vast number (millions) of antigen receptor sequences using high-throughput sequencing techniques has dramatically changed our possibilities to explore immune (BCR) repertoires. Clonally-related sequences in a BCR repertoire descend from a common ancestor and present the same V(D)J rearrangement, but they may differ due to the accumulation of SHM, making their automatic clonal grouping challenging. Clonal relationships can be computationally identified from a large set of IGH sequences using a clustering approach. Most clonal lineage grouping methods automatically separate sequences into clonal groups based on their similarities or distances, considering the whole sequence and (or) junction regions. Commonly, their clustering algorithms are based on only one criterion that minimizes intra-clonal distances. However, such a single measure often does not capture the different aspects of repertoires, while considering multiple objectives should be more appropriate and natural. Here we propose MobiLLe, a Multi-Objective Based clustering for Inferring BCR clonal lineages from high-throughput B cell rEpertoire data. The method first produces initial groups containing sequences with identical IGHV and IGHJ germline annotations and more than s% of amino acid identity on CDR3 regions (default s = 70%). Next, groups can be merged if it leads to lower intra-clonal and higher inter-clonal diversity. Thus MobiLLe optimizes two objective functions that continually evaluate clonal lineages' consistency until no improvement is observed in their cohesion or separation.

We validated our method on synthetic data that simulated three types of immune repertoires (monoclonal, oligoclonal, and polyclonal) with different SHM rates. We first carried out an extensive parameter optimization to estimate the influence of the MobiLLe parameter on its performance. We detected several parameter configurations that can achieve the best performance on the repertoires with different characteristics, Section 3.1. With default parameters, MobiLLe inferred clonal relationships with very high accuracy, detecting clonal memberships and precisely reconstructed repertoire structures. The simulations showed that existing algorithms frequently over-split clonal lineages, separating sequences belonging to the same B cell lineage and originating from a common ancestral. High SHM rates impact the clustering accuracy of most methods; they achieved lower performance on simulated repertoires that contain divergent sequences, mainly on monoclonal samples. On the other hand, MobiLLe was very stable, keeping an excellent performance independently of SHM rates and repertoire types.

To evaluate performance on experimental data, we generated three artificial monoclonal repertoires by mixing sequences from a pure B cell lineage with sequences from polyclonal repertoires. Then, we measured the ability of the existing clonal lineage grouping tools and MobiLLe to cluster members of the most abundant group (the pure B cell lineage). MobiLLe accurately grouped sequences from the B cell lineage and separated those from the polyclonal background in different clusters. We observed that MobiLLe performed the minimum number of splits and detected fewer false positives when compared with other tools. In addition, our method allows defining rules to group sequences with different CDR3 lengths. It can be helpful since SHM also introduce insertions and deletions (indels) in the junction region at a low-frequency [42, 43]. It was the case of the AMR1 benchmark where we observed indels of a

tyrosine residue at the IGHD-IGHJ junction region. Thanks to this flexibility MobiLLe reconstructed the clonal lineage of the AMR1 benchmark better. Other rules or exceptions can be easily included within MobiLLe framework upon specific needs in the clustering process.

After validating our method on simulated and experimental-based benchmarks, we applied it to experimental datasets where clonal lineage groups are unknown. We considered 26 experimental repertoires from different sources, sampled from diverse tissues, representing several human clinical conditions such as leukemia (CLL) and infection (COVID-19). Nine repertoires were sequenced at the Pitié-Salpêtrière hospital (Paris-France), three of them contained CLL cells, and six were considered non-clonal (polyclonal). These repertoires were used to check if MobiLLe could accurately detect the clonality status in patients with lymphoid malignancies and distinguish them from healthy donors. MobiLLe was able to estimate repertoires' clonality and provide additional information about clonal distribution for the nine considered repertoires with results in agreement with those obtained by conventional clonality assessment techniques (PCR + Genescan) for all investigated samples. Thus, MobiLLe, can predict clonality assignment and distinguish between clonal and non-clonal B cell populations. In addition, we measured the disequilibrium of a repertoire using the Gini index applied to cluster size distribution. We observed that monoclonal repertoires presented the highest Gini indices, indicating a disequilibrium in the clonal population. On the other hand, non-clonal repertoires presented lower Gini indices, showing a more diversified dataset.

Three experimental repertoires were used to compare MobiLLe with existing clonal lineage grouping tools. MobiLLe clonal distributions were often closer to those of Partis and SCOPe, the most performing tools. The pairwise evaluation confirmed that the clustering results of Partis and SCOPe were closer to MobiLLe with a F-score superior to 0.93 for all analyzed repertoires. However, MobiLLe is computationally more efficient than Partis and does not require an optimized distance threshold as SCOPe, which can be time-consuming, especially when analyzing monoclonal repertoires.

In order to validate our method on IGH sequences originating from other sources, we applied it to two independent datasets. The first one contained five B-cell repertoire samples originating from several lymphoproliferative diseases, previously published in [34]. MobiLLe reproduced the results of that study, identifying clonal populations in patients with different lymphoproliferative disorders. The second dataset included 12 repertoires selected from a study that identified B cell oligoclonal expansions in severe/moderate COVID-19 patients. MobiLLe achieved the same conclusion, detecting perturbations in the repertoires associated with the severity of the disease.

Accurate identification of clonal lineage members is essential for many repertoire analyzes. However, the design and development of BCR clonal lineage grouping methods present several challenges, such as determining sequence similarities, choosing threshold distances, and maximizing computational efficiency. We showed that MobiLLe clustering algorithm could identify clonal lineages with high accuracy and low runtime. We believe that our composed distance that analyses each sequence part is more appropriate and yields better results than previous definitions. A fixed distance threshold choice seems counterbalanced by our multi-objective optimization approach, which optimizes the trade-off between intra-clonal cohesion and inter-clonal separability.

Finally, MobiLLe is not as computationally demanding as other methods that compute likelihood-based inference or optimized distance thresholds. A possible improvement is to continue merging clusters when their combined uniformity (Eq 2) minimizes intra-clonal distances and maximizes inter-clonal distances. This could improve the performance when high pre-clustering thresholds are chosen. A possible extension is to adapt the method for

grouping epitope-specific T cell receptor sequences; could be an alternative to powerful, but time-consuming methods based on deep neural networks [44].

## 5 Conclusion

BCR lineage clonal grouping is at the core of BCR repertoire analysis. All downstream investigations, such as repertoire diversity estimation and intra-clonal analysis, depend on the correct grouping of BCR sequences. Several BCR clonal lineage grouping methods have been proposed, but the most performing tools are either time-consuming or unstable when higher divergent repertoires are analyzed. MobiLLe is a fast and accurate tool for BCR lineage clonal grouping with low runtime and memory requirements and does not require a training process or hyper-parameter optimization. It can easily be applied to (very) large-scale experimental repertoires, providing useful plots that could help interpret BCR lineage groups. An implementation of MobiLLe is freely available in Github https://github.com/julibinho/MobiLLe.

## Supporting information

**S1 Table. Initial clonal size distribution for the three types of simulated repertoires.** The clonal group sizes were defined to guarantee that monoclonal repertoires will contain a major clonal lineage with at least 70% of sequences, oligoclonal repertoires will include two clonal lineages representing 14% and 9% of sequences and any clonal group in polyclonal repertoires will not contain more than 5% of sequences.
(XLSX)

**S2 Table. Performance on the monoclonal repertoire M16.** The sample was generated with $\lambda_0 = 0.16$, number of sequences = 958 and number of expected clusters = 34.
(XLSX)

**S3 Table. Performance on the oligoclonal repertoire O16.** The sample was generated with $\lambda_0 = 0.16$, number of sequences = 1014 and number of expected = 43.
(XLSX)

**S4 Table. Performance on the polyclonal repertoire P16.** The sample was generated with $\lambda_0 = 0.16$, number of sequences = 968 and number of expected = 44.
(XLSX)

**S5 Table. Performance on the monoclonal repertoire M26.** The sample was generated with $\lambda_0 = 0.26$, number of sequences = 659 and number of expected = 33.
(XLSX)

**S6 Table. Performance on the oligoclonal repertoire O26.** The sample was generated with $\lambda_0 = 0.26$, number of sequences = 958 and number of expected = 43.
(XLSX)

**S7 Table. Performance on the polyclonal repertoire P26.** The sample was generated with $\lambda_0 = 0.26$, number of sequences = 964 and number of expected = 44.
(XLSX)

**S8 Table. Performance on the monoclonal repertoire M36.** The sample was generated with $\lambda_0 = 0.36$, number of sequences = 924 and number of expected = 35.
(XLSX)

**S9 Table. Performance on the oligoclonal repertoire O36.** The sample was generated with $\lambda_0$ = 0.36, number of sequences = 991 and number of expected = 40.
(XLSX)

**S10 Table. Performance on the polyclonal repertoire P36.** The sample was generated with $\lambda_0$ = 0.36, number of sequences = 897 and number of expected = 42.
(XLSX)

**S11 Table. Performance on the monoclonal repertoire M46.** The sample was generated with $\lambda_0$ = 0.46, number of sequences = 952 and number of expected = 35.
(XLSX)

**S12 Table. Performance on the oligoclonal repertoire O46.** The sample was generated with $\lambda_0$ = 0.46, number of sequences = 1016 and number of expected = 43.
(XLSX)

**S13 Table. Performance on the polyclonal repertoire P46.** The sample was generated with $\lambda_0$ = 0.46, number of sequences = 952 and number of expected = 43.
(XLSX)

**S14 Table. Comparison of MobiLLe with four different clonal lineage grouping methods on three experimental repertoires: $I_1$, $I_6$, and $I_8$, by using the pairwise evaluation method.**
(XLSX)

**S15 Table. Comparison of MobiLLe with four different clonal lineage grouping methods on three experimental repertoires: $I_1$, $I_6$, and $I_8$, by using the closeness evaluation method.**
(XLSX)

**S16 Table. Comparison of MobiLLe computational time (in seconds) with four different clonal lineage grouping methods on three experimental repertoires $I_1$, $I_6$, and $I_8$.**
(XLSX)

**S1 Fig. Different levels of grouping clonally-related sequences in a BCR repertoire.** The first level represents the entire set of sequences without any grouping. The second level represents B cell lineages. Sequences within a clonal lineage have the same V(D)J rearrangement and evolved from a common ancestor. The third level groups clonally-related sequences with identical CDR3 amino acid content, forming a so-called sub-clone. The fourth level groups identical nucleotide sequences within a given sub-clone, termed as clonotype level.
(TIFF)

**S2 Fig. IGHV/J gene usage distribution of the AMR1 repertoire compared to the polyclonal background.** (A) The IGHV gene usage. (B) The IGHJ gene usage. The polyclonal background is in gray and the gene segment of AMR1 is shown in blue.
(TIFF)

**S3 Fig. IGHV/J gene usage distribution of the AMR2 repertoire compared to the polyclonal background.** (A) The IGHV gene usage. (B) The IGHJ gene usage. The polyclonal background is in gray and the gene segment of AMR2 is shown in blue.
(TIFF)

**S4 Fig. IGHV/J gene usage distribution of the AMR3 repertoire compared to the polyclonal background.** (A) The IGHV gene usage. (B) The IGHJ gene usage. The polyclonal background is in gray and the gene segment of AMR3 is shown in blue.
(TIFF)

**S5 Fig. GeneScan profiles of human peripheral blood samples.** IGH-VDJ rearrangements were amplified using conventional methods and PCR products were further analyzed by capillary electrophoresis. (A-C) Samples from individuals with monoclonal B cell malignancy: monoallelic profile (A and C) or biallelic profile (B); (D-I) non-malignant samples: regular polyclonal profile (D, E, G, H, I) or irregular polyclonal profile (F).
(TIFF)

**S6 Fig. Clustering performance measurements.** A) Pairwise B) Closeness.
(TIFF)

**S7 Fig. Circle representation of the 100 most abundant clonal lineages for the experimental dataset produced at the Pitié-Salpêtrière hospital.** Each circle symbolizes a clonal lineage, and the circle area its abundance. The ordinate represents cluster uniformity Eq 2, while the abscissa the clonal lineage abundance in %. (A) $I_1$, (B) $I_2$, (C) $I_3$, (D) $I_4$, (E) $I_5$, (F) $I_6$, (G) $I_7$, (H) $I_8$, and (I) $I_9$.
(TIFF)

**S8 Fig. Repertoire of individual 1.** A) Circle representation of clonal group abundance. Each circle symbolizes a clonal lineage, and its area is proportional to the clonal group abundance. B) Number of sequences in each group, all clonal lineages are represented, vertical axis is in log scale. C) Lorenz curve and Gini coefficient. A Lorenz curve shows the graphical representation of clonal inequality. On the horizontal axis, it plots the cumulative fraction of total clonal lineages when ordered from the least to the most abundant; on the vertical axis, it shows the cumulative fraction of sequences. D) Clonal size distribution (percentage) of the 100 most abundant clonal lineages.
(TIFF)

**S9 Fig. Repertoire of individual 2.** A) Circle representation of clonal group abundance. Each circle symbolizes a clonal lineage, and its area is proportional to the clonal group abundance. B) Number of sequences in each group, all clonal lineages are represented, vertical axis is in log scale. C) Lorenz curve and Gini coefficient. A Lorenz curve shows the graphical representation of clonal inequality. On the horizontal axis, it plots the cumulative fraction of total clonal lineages when ordered from the least to the most abundant; on the vertical axis, it shows the cumulative fraction of sequences. D) Clonal size distribution (percentage) of the 100 most abundant clonal lineages.
(TIFF)

**S10 Fig. Repertoire of individual 3.** A) Circle representation of clonal group abundance. Each circle symbolizes a clonal lineage, and its area is proportional to the clonal group abundance. B) Number of sequences in each group, all clonal lineages are represented, vertical axis is in log scale. C) Lorenz curve and Gini coefficient. A Lorenz curve shows the graphical representation of clonal inequality. On the horizontal axis, it plots the cumulative fraction of total clonal lineages when ordered from the least to the most abundant; on the vertical axis, it shows the cumulative fraction of sequences. D) Clonal size distribution (percentage) of the 100 most abundant clonal lineages.
(TIFF)

**S11 Fig. Repertoire of individual 4.** A) Circle representation of clonal group abundance. Each circle symbolizes a clonal lineage, and its area is proportional to the clonal group abundance. B) Number of sequences in each group, all clonal lineages are represented, vertical axis is in log scale. C) Lorenz curve and Gini coefficient. A Lorenz curve shows the graphical representation of clonal inequality. On the horizontal axis, it plots the cumulative fraction of total clonal

lineages when ordered from the least to the most abundant; on the vertical axis, it shows the cumulative fraction of sequences. D) Clonal size distribution (percentage) of the 100 most abundant clonal lineages.

(TIFF)

**S12 Fig. Repertoire of individual 5.** A) Circle representation of clonal group abundance. Each circle symbolizes a clonal lineage, and its area is proportional to the clonal group abundance. B) Number of sequences in each group, all clonal lineages are represented, vertical axis is in log scale. C) Lorenz curve and Gini coefficient. A Lorenz curve shows the graphical representation of clonal inequality. On the horizontal axis, it plots the cumulative fraction of total clonal lineages when ordered from the least to the most abundant; on the vertical axis, it shows the cumulative fraction of sequences. D) Clonal size distribution (percentage) of the 100 most abundant clonal lineages.

(TIFF)

**S13 Fig. Repertoire of individual 6.** A) Circle representation of clonal group abundance. Each circle symbolizes a clonal lineage, and its area is proportional to the clonal group abundance. B) Number of sequences in each group, all clonal lineages are represented, vertical axis is in log scale. C) Lorenz curve and Gini coefficient. A Lorenz curve shows the graphical representation of clonal inequality. On the horizontal axis, it plots the cumulative fraction of total clonal lineages when ordered from the least to the most abundant; on the vertical axis, it shows the cumulative fraction of sequences. D) Clonal size distribution (percentage) of the 100 most abundant clonal lineages.

(TIFF)

**S14 Fig. Repertoire of individual 7.** A) Circle representation of clonal group abundance. Each circle symbolizes a clonal lineage, and its area is proportional to the clonal group abundance. B) Number of sequences in each group, all clonal lineages are represented, vertical axis is in log scale. C) Lorenz curve and Gini coefficient. A Lorenz curve shows the graphical representation of clonal inequality. On the horizontal axis, it plots the cumulative fraction of total clonal lineages when ordered from the least to the most abundant; on the vertical axis, it shows the cumulative fraction of sequences. D) Clonal size distribution (percentage) of the 100 most abundant clonal lineages.

(TIFF)

**S15 Fig. Repertoire of individual 8.** A) Circle representation of clonal group abundance. Each circle symbolizes a clonal lineage, and its area is proportional to the clonal group abundance. B) Number of sequences in each group, all clonal lineages are represented, vertical axis is in log scale. C) Lorenz curve and Gini coefficient. A Lorenz curve shows the graphical representation of clonal inequality. On the horizontal axis, it plots the cumulative fraction of total clonal lineages when ordered from the least to the most abundant; on the vertical axis, it shows the cumulative fraction of sequences. D) Clonal size distribution (percentage) of the 100 most abundant clonal lineages.

(TIFF)

**S16 Fig. Repertoire of individual 9.** A) Circle representation of clonal group abundance. Each circle symbolizes a clonal lineage, and its area is proportional to the clonal group abundance. B) Number of sequences in each group, all clonal lineages are represented, vertical axis is in log scale. C) Lorenz curve and Gini coefficient. A Lorenz curve shows the graphical representation of clonal inequality. On the horizontal axis, it plots the cumulative fraction of total clonal lineages when ordered from the least to the most abundant; on the vertical axis, it shows the

cumulative fraction of sequences. D) Clonal size distribution (percentage) of the 100 most abundant clonal lineages.
(TIFF)

**S17 Fig. Circle representation of the top 100 ranked clonal lineages for the experimental dataset containing repertoires from patients with different lymphoproliferative diseases and a healthy donor.** Each circle symbolizes a clonal lineage, and the circle area is proportional to clonal group abundance. The ordinate represents cluster uniformity (Eq 2), while the abscissa the clonal lineage abundance in %. (A) HD, (B) P3-FL-SLL, (C) P4-C/SLL, (D)P5-BM, and (E) P5-L. Report to Table 3 for repertoires' properties and individuals' labels.
(TIFF)

**S18 Fig. Circle representation of the the top 100 ranked clonal lineages for the experimental dataset containing repertoires from patients with severe COVID-19.** Each circle symbolizes a clonal lineage, and the circle area is proportional to clonal group abundance. The ordinate represents cluster uniformity (Eq 2), while the abscissa the clonal lineage abundance in %. (A) S20, (B) S21, (C) S22, (D) S23, (E) S24, and (F) S26. Report to Table 4 for repertoires' properties and individuals' labels.
(TIFF)

**S19 Fig. Circle representation of the the top 100 ranked clonal lineages for the experimental dataset containing repertoires from patients with moderate COVID-19 and healthy donors.** Each circle symbolizes a clonal lineage, and the circle area is proportional to clonal group abundance. The ordinate represents cluster uniformity (Eq 2), while the abscissa the clonal lineage abundance in %. (A) H3, (B) H4, (C) H8, (D) M5, (E) M6, and (F) M7. Report to Table 4 for repertoires' properties and individuals' labels.
(TIFF)

## Author Contributions

**Conceptualization:** Juliana Silva Bernardes.

**Data curation:** Anne Langlois De Septenville.

**Investigation:** Juliana Silva Bernardes.

**Methodology:** Nika Abdollahi, Juliana Silva Bernardes.

**Project administration:** Juliana Silva Bernardes.

**Software:** Nika Abdollahi, Lucile Jeusset, Hugues Ripoche, Juliana Silva Bernardes.

**Supervision:** Frédéric Davi.

**Validation:** Nika Abdollahi, Lucile Jeusset, Anne Langlois De Septenville, Frédéric Davi, Juliana Silva Bernardes.

**Visualization:** Nika Abdollahi, Lucile Jeusset, Hugues Ripoche.

**Writing – original draft:** Juliana Silva Bernardes.

**Writing – review & editing:** Nika Abdollahi, Frédéric Davi.

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
