## [Decision Letter · Decision Letter 0]

23 Feb 2022

Dear Dr Bernardes,

Thank you very much for submitting your manuscript "A multi-objective based clustering for inferring BCR clones from high-throughput B-cell repertoire data" for consideration at PLOS Computational Biology.

As with all papers reviewed by the journal, your manuscript was reviewed by members of the editorial board and by several independent reviewers. In light of the reviews (below this email), we would like to invite the resubmission of a significantly-revised version that takes into account the reviewers' comments.

We cannot make any decision about publication until we have seen the revised manuscript and your response to the reviewers' comments. Your revised manuscript is also likely to be sent to reviewers for further evaluation.

Sincerely,

Philipp Altrock

Guest Editor

PLOS Computational Biology

Jason Haugh

Deputy Editor

PLOS Computational Biology

Reviewer's Responses to Questions

**Comments to the Authors:**

Reviewer #1: The overall goal of this manuscript is to provide an algorithm whereby, before BCR diversity and sub categories are established, very closely related sequences are grouped. This approach would presumably add more power to the final goal of mapping the gross clustering results or the distinctions onto a given target, for example, outcomes from a disease or infection. Overall, this goal is new and will be a very useful contribution to the field. (Algorithm 1 figure is particularly useful and clear.)

Here are some minor issues.

1. Authors seem to go back and forth between clonal grouping and identical CDR3 junctions versus similarity of CDR3 junctions with the same V’s and J’s. This seems to be particularly a problem in the Introduction. This reviewer believes the term clonal represents usually represents identity to most readers…Authors should consider reviewing their language, especially in the introduction and make a clear distinction between identical sequences, from V to CDR3 to J versus closely related sequences that may indeed be targeting the same epitope.

2. With regard to number 1 above, are the authors trying to indicate that a V-CDR3-J defines the clones and the somatic hyper-mutation divergence is what is the basis for “intra-clonal grouping”? Once again, if so, that may need to be written out more carefully. Except, this preceding consideration may not be accurate because authors refer to clonal grouping based on CDR3 lengths rather than exact amino acid sequences??

3. Exactly which kind of leukemia does this phrase refer to: “Three of these samples contained clonal leukemic cells”? Do these three leukemias represent a diagnosis of B-cell, acute lymphocytic leukemia?

4. The Fig. 3, 4 legends should be divided into A-F (or A-I) with explanations for each panel.

5. It really seems like at least one more experimental set should be evaluated. If not, the authors should discuss specific limitations. For example, have the authors considered the ireceptor.org repository or a collection from Adaptive Biotechnologies?

Reviewer #2: Abdollahi et al. developed a tool for BCR clustering based on multi-objective clustering approach. The performance showed seems promising based on simulations and comparisons with similar methods. The strength of the manuscript is that simulations are well planned, with very detailed results reported in the results section. The weakness of the manuscript is the method itself which seems trivial. Also the implementation (the software provided in the GitHub) has limited options for parameter tuning or customization. Overall, I think the manuscript can be improved by potentially addressing the following questions:

(1) There is no discussion regarding why this paper should only focus on BCR and why the same method (multi-objective) clustering cannot be applied to TCR clustering.

(2) Although many methods are compared, there is a lack of discussion or comparison with another popular method for multi-objective clustering based on the deep learning framework (such as auto encoder, e.g., Nature Communications 12: 1605 (2021))

(3) There are other distance metrics such as the geometric Isometry-based (implemented in GIANA) has shown to achieve much faster computational speed. Is this one possible option for this implementation?

(4) It would really strengthen the paper if authors present their results on real TCR reads (such as the BCR profile called from TCGA, published by Liu Group) or provide their tool as an online resource where users can upload their reads for fast clustering analysis.

Reviewer #3: General assessment of the work:

In this work, the authors present a clustering algorithm, MobiLLe, for B-cell immune repertoire datasets. It has multiple objectives and allows the refinement of clones by minimizing intraclonal distances and maximizing interclonal distances. They show MobiLLe’s performance on synthetic data produced using GCTree and on experimental data. They also compare MobiLLe's performance to other clustering algorithms. Overall, the authors show a promising, interesting algorithm, but I don’t know how it will work on most datasets without being able to handle singleton clones.

Major comments:

Equation 1 presents the distance between sequences. The authors stated the distance between V genes takes on a binary value whereas J genes do not. There are quite a few V genes which are very similar to each other (at least 90% similar in normalized Hamming distance using nucleotides). Moreover, some V genes are quite synonymous with one another, possibly differing in, say, only one amino acid. I have two questions. What motivated the choice for the binary measure for V gene distances? How does the performance change using a Levenshtein distance measure as is or one which weights changes in the framework and complementarity determining regions differently?

While the authors detail the properties of the monoclonal repertoires, it’s not clear to me what the polyclonal repertoire is composed of, i.e., how different is the background from the signal. Could the authors have a figure indicating, at the very least, how the V gene, J gene, and CDR3 distributions of the background repertoires appear? I'm imagining six plots. Three plots showing the statistics at the level of MoibLLe's clustering using lineage counts and three plots showing the statistics at the level of unique counts for each sequence (better yet, unique counts for each nonsingleton sequences if that abundance information is available). On these plots, could they please indicate the signal V and J genes and CDR3 length, so that everything is summarized on one figure.

How does MobiLLe compare to hierarchical clustering alone? While comparing to these other algorithms is useful, I don’t have a sense of how much refinement MobiLLe is performing compared to its initial step.

Fixed hierarchical clustering thresholds are typically chosen at 85% or 90% based on the bimodal features apparent from the normalized Hamming distance of nearest neighbors in a V gene, J gene, and CDR3 length bin. Because of this, I’m very curious how MobiLLe can be used with higher CDR3 similarity in the preclustering step. In practice, I would expect many singleton clones because B-cell immune repertoires are highly undersampled. Besides getting MobiLLe to work, how can the authors motivate choosing lower similarity thresholds either statistically or biologically? What are the impacts in analyses of precluding singleton clones from existing? Might the authors be able to implement this by the next iteration of reviews?

Minor comments:

Fig 4: Because so many circles overlap, what’s being presented is obscured. Is it possible to coarse grain the information being presented? Unless I don’t understand what’s being presented, could the authors simply present log PDF vs. log clone abundance with a histogram, line, or scatter plot?

Fig 5: I don’t know what the “agreeable” label means. Is this MobiLLe’s performance?

Fig S1: Axis labels? I don’t know what’s being shown.

Fig. S2: There is no T3 in the diagram. In the main text and in the equations, T3 shows up.

**Have the authors made all data and (if applicable) computational code underlying the findings in their manuscript fully available?**

Reviewer #1: Yes

Reviewer #2: Yes

Reviewer #3: **No: **As far as I can tell, the experimental data or artificial experimental datasets are not provided.

PLOS authors have the option to publish the peer review history of their article (what does this mean?). If published, this will include your full peer review and any attached files.

Reviewer #1: No

Reviewer #2: No

Reviewer #3: No
---

## [Decision Letter · Decision Letter 1]

26 Jun 2022

Dear Dr Bernardes,

Thank you very much for submitting your manuscript "A multi-objective based clustering for inferring BCR clonal lineages from high-throughput B cell repertoire data" for consideration at PLOS Computational Biology. As with all papers reviewed by the journal, your manuscript was reviewed by members of the editorial board and by several independent reviewers. The reviewers appreciated the attention to an important topic. Based on the reviews, we are very likely to accept this manuscript for publication, providing that you modify the manuscript according to the remaining recommendations by one of the reviewers.

Sincerely,

Philipp M Altrock, Ph.D.

Guest Editor

PLOS Computational Biology

Jason Haugh

Deputy Editor

PLOS Computational Biology

[LINK]

Reviewer's Responses to Questions

**Comments to the Authors:**

Reviewer #1: Authors successfully addressed all of my concerns.

Reviewer #2: All my questions are properly addressed.

Reviewer #3: Thank you to the authors for their revisions. The authors have addressed my concerns.

Minor comments:

Please read through your manuscript thoroughly. There are some typos that cause a lot of confusion. Thanks.

Algorithm 2: Is the b_i line necessary? b_i doesn’t appear to be used in this algorithm.

Fig. 2: Change S14 to S13 in panel B.

Fig. 11: Should the y-axis read “Gini index” or “abundance”?

Fig 13: Panel C’s y-axis should read “HCDR3 length [nt]” or something like that, no? Please make sure everything that is labeled “abundance” is actually abundance.

Line 780: “Fig S19 and S19” to “Fig S18 and S19”

Create an environment file for your package instead of listing what needs to be installed in the README.md: https://docs.conda.io/projects/conda/en/latest/user-guide/tasks/manage-environments.html#create-env-file-manually

**Have the authors made all data and (if applicable) computational code underlying the findings in their manuscript fully available?**

Reviewer #1: Yes

Reviewer #2: Yes

Reviewer #3: Yes

PLOS authors have the option to publish the peer review history of their article (what does this mean?). If published, this will include your full peer review and any attached files.

Reviewer #1: No

Reviewer #2: No

Reviewer #3: No

Figure Files:

Data Requirements:

Reproducibility:

References:

---

## [Editor Report · Decision Letter 2]

18 Jul 2022

Dear Dr Bernardes,

We are pleased to inform you that your manuscript 'A multi-objective based clustering for inferring BCR clonal lineages from high-throughput B cell repertoire data' has been provisionally accepted for publication in PLOS Computational Biology.

Also, please address the last round of minor comments by one of the reviewers.

Best regards,

Philipp M Altrock, Ph.D.

Guest Editor

PLOS Computational Biology

Jason Haugh

Deputy Editor

PLOS Computational Biology

Please address these final minor comments from one of the referees:

Minor comments:

Please read through your manuscript thoroughly. There are some typos that cause a lot of confusion. Thanks.

Algorithm 2: Is the b_i line necessary? b_i doesn’t appear to be used in this algorithm.

Fig. 2: Change S14 to S13 in panel B.

Fig. 11: Should the y-axis read “Gini index” or “abundance”?

Fig 13: Panel C’s y-axis should read “HCDR3 length [nt]” or something like that, no? Please make sure everything that is labeled “abundance” is actually abundance.

Line 780: “Fig S19 and S19” to “Fig S18 and S19”

Create an environment file for your package instead of listing what needs to be installed in the README.md: https://docs.conda.io/projects/conda/en/latest/user-guide/tasks/manage-environments.html#create-env-file-manually

---

## [Editor Report · Acceptance letter]

19 Aug 2022

PCOMPBIOL-D-21-02237R2 

A multi-objective based clustering for inferring BCR clonal lineages from high-throughput B cell repertoire data

Dear Dr Bernardes,

I am pleased to inform you that your manuscript has been formally accepted for publication in PLOS Computational Biology. Your manuscript is now with our production department and you will be notified of the publication date in due course.

With kind regards,

Agnes Pap
